# Neural varifolds: an aggregate representation for quantifying the geometry of point clouds

**Juheon Lee**[*]                                         *juheon.lee.626@gmail.com*
*Independent Researcher*

**Xiaohao Cai**                                              *X.Cai@soton.ac.uk*
*School of Electronics and Computer Science, University of Southampton*

**Carola-Bibiane Schönlieb**                                     *cbs31@cam.ac.uk*
*Department of Applied Mathematics and Theoretical Physics, University of Cambridge*

**Simon Masnou**                                  *masnou@math.univ-lyon1.fr*
*Institut Camille Jordan, Université Claude Bernard Lyon 1*

**Reviewed on OpenReview:** *https://openreview.net/forum?id=PO2hoA7vln*

## Abstract

Point clouds are popular 3D representations for real-life objects (such as in LiDAR and Kinect) due to their detailed and compact representation of surface-based geometry. Recent approaches characterise the geometry of point clouds by bringing deep learning based techniques together with geometric fidelity metrics such as optimal transportation costs (e.g., Chamfer and Wasserstein metrics). In this paper, we propose a new surface geometry characterisation within this realm, namely a neural varifold representation of point clouds. Here, the surface is represented as a measure/distribution over both point positions and tangent spaces of point clouds. The varifold representation quantifies not only the surface geometry of point clouds through the manifold-based representation, but also subtle geometric consistencies on the surface due to the combined product space. This study proposes neural varifold algorithms to compute the varifold norm between two point clouds using neural networks on point clouds and their neural tangent kernel representations. The proposed neural varifold is evaluated on three different sought-after tasks – shape matching, few-shot shape classification, and shape reconstruction. Detailed evaluation and comparison to the state-of-the-art methods demonstrate that the proposed versatile neural varifold is superior in shape matching and few-shot shape classification, and is competitive for shape reconstruction. The public code is available at `https://github.com/jl626/neural_varifold`.

## 1 Introduction

Point clouds are preferred in applications including computer graphics, autonomous driving, robotics, and augmented reality. However, manipulating/editing point clouds data in its raw form is rather cumbersome. Neural networks have made breakthroughs in a wide variety of fields ranging from natural language processing to computer vision. Point cloud data in general lack underlying grid structures. As a result, convolution operations on point cloud data require special techniques including voxelisation (Deng et al., 2021; Shi et al., 2020; Choy et al., 2019), graph representations (Bruna et al., 2014; Bronstein et al., 2017; Wang et al., 2019) or pointwise convolutions (Qi et al., 2017a;b; Thomas et al., 2019). Geometric deep learning and its variants have addressed technical problems of translating neural networks on point cloud data (Bronstein et al., 2017). With advanced graph theory and harmonic analysis, convolutions on point cloud data can be defined in the

---

[*]This research is my personal endeavor and is unrelated to my current job.

context of spectral (Bruna et al., 2014; Defferrard et al., 2016) or spatial (Monti et al., 2017; Wang et al., 2019) domains. Although geometric deep learning on point clouds has successfully achieved top performance in shape classification and segmentation tasks, capturing subtle changes in 3D surface remains challenging due to the unstructured and non-smooth nature of point clouds. A possible direction to learn subtle changes on 3D surface adopts some concepts developed in the field of theoretical geometric analysis. In other words, deep learning architectures might be improved by incorporating theoretical knowledge from geometric analysis. In this work, we introduce concepts borrowed from geometric measure theory, where the representation of shapes as measures or distributions has been instrumental.

Geometric measure theory has been actively investigated by mathematicians; however, its technicality may have hindered its popularity and use in many applications. Geometric measure-theoretic concepts have recently been introduced to measure shape correspondence in non-rigid shape matching (Vaillant & Glaunès, 2005; Charon & Trouvé, 2013; Hsieh & Charon, 2021) and curvature estimation (Buet et al., 2017; 2022). We introduce the theory of varifolds to improve learning representation of 3D point clouds. An oriented $d$-varifold is a measure over point positions and oriented tangent $k$-planes, i.e., a measure on the Cartesian product space of $\mathbb{R}^n$ and the oriented Grassmannian manifold $\tilde{G}(d, n)$. Varifolds can be viewed as generalisations of $d$-dimensional smooth shapes in Euclidean space $\mathbb{R}^n$. The varifold structure helps to better differentiate not only the macro-geometry of the surface through the manifold-based representation, but also the subtle singularities in the surface due to the combined product space. Varifolds provide representations of general surfaces without parameterization. They not only can represent consistently point clouds that approximate surfaces in 3D, but are also scalable to arbitrary surface discretisation (e.g., meshes). In this study, we use varifolds to analyse and quantify the geometry of point clouds.

**Our contributions:**

- Introduce the notion of neural varifold as a learning representation of point clouds. Varifold representation of 3D point clouds coupling space position and tangent planes can provide both theoretical and practical analyses of the surface geometry.

- Propose two algorithms to compute the varifold norm between two point clouds using neural networks on point clouds and their neural tangent kernel representations. The reproducing kernel Hilbert space of the varifold is computed by the product of two neural tangent kernels of positional and Grassmannian features of point clouds. The neural varifold can take advantage of the expressive power of neural networks as well as the varifold representation of point clouds.

- Apply the usage of neural varifold in evaluating shape similarity between point clouds on various tasks including shape matching, few-shot shape classification and shape reconstruction.

## 2 Related works

**Geometric deep learning on point clouds.** PointNet is the first pioneering work on point clouds. It consists of a set of fully connected layers followed by symmetric functions to aggregate feature representations. In other words, PointNet is neural networks on a graph without edge connections. In order to incorporate local neighbourhood information with PointNet, PointNet++ (Qi et al., 2017b) applied PointNet to individual patches of the local neighbourhood, and then stacked them together. PointCNN (Li et al., 2018) further refined the PointNet framework with hierarchical $\mathcal{X}$-Conv which calculates inner products of $\mathcal{X}$-transformation and convolution filters of point clouds. Dynamic graph CNN (DGCNN) (Wang et al., 2019) adopted the graph neural network framework to incorporate local neighbourhood information by applying convolutions over the graph edges and dynamically updating graph for each layer. Furthermore, the tangent convolution architecture (Tatarchenko et al., 2018) incorporated 3D surface geometry by projecting point clouds on local tangent planes, and then applying convolution filters.

**Shape similarity measures.** Quantifying the shape similarity between point clouds or other 3D modality has been an active area of research. In the literature, Chamfer distance (CD) and Earth Mover's Distance (EMD) are two most popular forms in the context of point cloud analysis (Achlioptas et al., 2018; Wu

et al., 2021). CD is a nearest-neighbor-based metric. Although it is computationally efficient, its accuracy drops significantly when the densities of the two point clouds differ greatly or when some subsets are much denser than others. EMD, on the other hand, is based on optimal transport theory and involves solving an optimization problem to construct a mapping from one point cloud to another (Achlioptas et al., 2018; Nguyen et al., 2021). Although EMD is more robust to noise and variations in density, it is computationally more expensive. In the context of medical imaging, EMD and CD lack desirable mathematical properties — such as guaranteeing diffeomorphic mappings — that are important for preserving the biological structure of organs and tissues. As a result, the Large Deformation Diffeomorphic Metric Mapping (LDDMM) framework has gained more attention (Dupuis et al., 1998; Vaillant & Glaunès, 2005; Charon & Trouvé, 2013). Although LDDMM provides biologically meaningful and topology-preserving transformations, it is computationally intensive, often requiring the solution of complex partial differential equations or high-dimensional optimization problems. This makes it less efficient and less suitable for general-purpose applications.

**Varifolds.** Geometric measure theory provides various tools for understanding, characterising and analysing surface geometry in various contexts, e.g., currents (Vaillant & Glaunès, 2005), varifolds (Charon & Trouvé, 2013; Buet et al., 2017; 2022) or normal cycles (Roussillon & Glaunès, 2019). Despite their potential use for many applications, few studies have explored real-world applications of varifolds in the context of non-rigid surface registration (Charon & Trouvé, 2013).

## 3 Varifold representations for point clouds

The notion of varifold was introduced in geometric measure theory in the context of the Plateau problem, that is, finding a minimal surface spanning a given closed curve in $\mathbb{R}^3$ (Almgren, 1964; Allard, 1972; Simon, 1983). Intuitively, the concept of varifold extends the notion of differentiable manifold by replacing the requirement for differentiability with a condition of rectifiability, or even weaker conditions (Menne, 2017). This modification enables the representation of more complex surfaces, including those with singularities and even point clouds (Menne, 2017; Buet et al., 2017; 2022). A general oriented $d$-varifold in an open set $\Omega \subset \mathbb{R}^n$ is a Radon measure on the product space of $\Omega$ with the oriented Grassmannian $\tilde{G}(d, n)$. For example, given a point cloud $\{x_i\}_{i=1}^N$ in $\mathbb{R}^n$ and a collection of oriented $d$-planes $\{T_i\}_{i=1}^N$, an oriented $d$-varifold naturally associated with the pairs $(x_i, T_i)$, $i = 1 \cdots N$, is given by $V = \sum_{i=1}^N \delta_{x_i} \otimes \delta_{T_i}$.

Various metrics and topologies can be defined in the space of varifolds. The mass distance is a possible choice for a metric:

$$d_{\text{mass}}(\mu, \nu) = \sup \left\{ \left| \int_{\Omega \times \tilde{G}(d,n)} \phi \mathrm{d}\mu - \int_{\Omega \times \tilde{G}(d,n)} \phi \mathrm{d}\nu \right|, \ \phi \in C_0(\Omega \times \tilde{G}(d,n)), \|\phi\|_\infty \leq 1 \right\}. \tag{1}$$

However, the mass distance is not well suited for point clouds. For example, the distance between varifolds associated with the Dirac masses $\delta_\varepsilon$ and $\delta_0$ remains bounded away from 0 as it is always possible to find a test function $\phi$ such that $|\phi(0) - \phi(\varepsilon)| = 2$, no matter how close $\varepsilon$ is to 0. The 1-Wasserstein distance is not a more suitable choice in our context since it cannot compare two varifold measures with different mass. For example, the 1-Wasserstein distance between varifolds associated with Dirac masses $(1 + \varepsilon)\delta_0$ and $\delta_0$ is infinite for all $\varepsilon \neq 0$. The bounded Lipschitz distance (or flat distance) has, in contrast, good properties; we refer for more details to Piccoli & Rossi (2016) and the references therein.

**Definition 3.1** (Bounded Lipschitz distance)**.** Being $\mu$ and $\nu$ two varifolds on a locally compact metric space $(X, d)$, we define

$$d_{\text{BL}}(\mu, \nu) = \sup \left\{ \left| \int_{\Omega \times \tilde{G}(d,n)} \phi \mathrm{d}\mu - \int_{\Omega \times \tilde{G}(d,n)} \phi \mathrm{d}\nu \right|, \ \phi \in C_0^1(\Omega \times \tilde{G}(d,n)), \ \|\phi\|_{\text{Lip}} \leq 1, \|\phi\|_\infty \leq 1 \right\}. \tag{2}$$

Although the bounded Lipschitz distance is a very suitable theoretical tool for comparing varifolds, in practice, there is no straightforward way to evaluate it numerically. Instead, the Reproducing Kernel Hilbert Space (RKHS) approach provides effective numerical tools for evaluating and comparing varifolds (Charon & Trouvé, 2013; Hsieh & Charon, 2021). In particular, the following proposition is useful.

**Proposition 3.2.** (Hsieh & Charon, 2021). *Let $k_{\text{pos}}$ and $k_G$ be continuous positive definite kernels on $\mathbb{R}^n$ and $\tilde{G}(d, n)$, respectively. Assume also that for any $x \in \mathbb{R}^n$, $k_{\text{pos}}(x, \cdot) \in C_0(\mathbb{R}^n)$. Then $k_{\text{pos}} \otimes k_G$ is a positive definite kernel on $\mathbb{R}^n \times \tilde{G}(d, n)$, and the RKHS $W$ associated with $k_{\text{pos}} \otimes k_G$ is continuously embedded in $C_0(\mathbb{R}^n \times \tilde{G}(d, n))$, i.e., there exists $c_W > 0$ such that for any $\phi \in W$, we have $\|\phi\|_\infty < c_W \|\phi\|_W$.*

Let $\tau_W : W \mapsto C_0(\mathbb{R}^n \times \tilde{G}(d, n))$ be a continuous embedding given by Proposition 3.2. Varifolds can be viewed, through the adjoint $\tau_W^*$, as elements of the dual Hilbert space $W^*$, and the Hilbert norm of $W^*$ provides a pseudo-metric (as $\tau_W^*$ is not injective in general): given two varifolds $\mu, \nu$,

$$d_{W^*}(\mu, \nu)^2 = \|\mu - \nu\|_{W^*}^2 = \|\mu\|_{W^*}^2 - 2\langle \mu, \nu \rangle_{W^*} + \|\nu\|_{W^*}^2 . \tag{3}$$

where

$$\langle \mu, \nu \rangle_{W^*} = \int_{(\mathbb{R}^n \times \tilde{G}(d, n))^2} k_{\text{pos}}(x, x') \, k_G(T, T') \, d\mu(x, T) \, d\nu(x', T').$$

This pseudo-metric provides an effective way to compare varifolds by separating the positional and Grassmannian components. One can derive a bound with respect to $d_{\text{BL}}$ if we further assume that $W$ is continuously embedded into $C_0^1(\mathbb{R}^n \times \tilde{G}(d, n))$ (Charon & Trouvé, 2013):

$$\|\mu - \nu\|_{W^*} = \sup_{\phi \in W, \|\phi\|_W \leq 1} \int_{\mathbb{R}^n \times \tilde{G}(d, n)} \phi \, d(\mu - \nu) \leq c_W d_{BL}(\mu, \nu).$$

**Neural tangent kernel.** Recent advances in neural network theory have revealed a link between kernel methods and over-parameterized neural networks (Jacot et al., 2018; Arora et al., 2019). Given training data pairs $\{x_i, y_i\}_{i=1}^M$, where $x_i \in \mathbb{R}^{d_0}$ and $y_i \in \mathbb{R}$, let $f(\theta; x_i)$ be a fully connected neural network with $L$-hidden layers with inputs $x_i$ and parameters $\theta = \{W^{(0)}, b^{(0)}, \cdots, W^{(L)}, b^{(L)}\}$. Let $d_h$ be the width of the neural network for each layer $h$. The neural network function $f$ can be written recursively as

$$f^{(h)}(x) = W^{(h)} g^{(h)}(x) + b^{(h)}, \quad g^{(h+1)}(x) = \varphi(f^{(h)}(x)), \quad h = 0, \ldots, L, \tag{4}$$

where $g^{(0)}(x) = x$ and $\varphi$ is a non-linear activation function.

Assume the weights $W^{(h)} \in \mathbb{R}^{d_{h+1} \times d_h}$ and bias $b^{(h)} \in \mathbb{R}^{d_h}$ at each layer $h$ are initialised with Gaussian distribution $W^{(h)} \sim \mathcal{N}(0, \sigma_\omega^2/d_h)$ and $b^{(h)} \sim \mathcal{N}(0, \sigma_b^2)$, respectively. Consider training a neural network by minimising the least square loss function

$$l(\theta) = \frac{1}{2} \sum_{i=1}^M (f(\theta; x_i) - y_i)^2. \tag{5}$$

Suppose the least square loss $l(\theta)$ is minimised with an infinitesimally small learning rate, i.e., $\frac{d\theta}{dt} = -\nabla l(\theta(t))$. Let $u(t) = (f(\theta(t); x_i))_{i \in [M]} \in \mathbb{R}^M$ be the neural network outputs on all $x_i$ at time $t$, and $y = (y_i)_{i \in [M]}$ be the desired output. Then $u(t)$ follows the evolution law

$$\frac{du}{dt} = -H(t)(u(t) - y), \tag{6}$$

where

$$H(t)_{ij} = \left\langle \frac{\partial f(\theta(t); x_i)}{\partial \theta}, \frac{\partial f(\theta(t); x_j)}{\partial \theta} \right\rangle. \tag{7}$$

If the width of the neural network at each layer goes to infinity, i.e. $d_h \to \infty$, with a fixed training set, then $H(t)$ remains unchanged. Under random initialisation of the parameters $\theta$, $H(0)$ converges in probability to a deterministic kernel $H^*$ – the "*neural tangent kernel*" (i.e. NTK) (Jacot et al., 2018). Indeed, with few known activation functions $\varphi$ (e.g. ReLU), the neural tangent kernel $H^*$ can be computed by a closed-form

solution recursively using a Gaussian process (Lee et al., 2017; Arora et al., 2019). For each layer $h$, the corresponding covariance function is defined as

$$\mathbf{\Sigma}^{(0)}(\boldsymbol{x}_i, \boldsymbol{x}_j) = \sigma_b^2 + \frac{\sigma_\omega^2}{d_0} \boldsymbol{x}_i \boldsymbol{x}_j^\top, \tag{8}$$

$$\mathbf{\Lambda}^{(h)}(\boldsymbol{x}_i, \boldsymbol{x}_j) = \begin{bmatrix} \mathbf{\Sigma}^{(h-1)}(\boldsymbol{x}_i, \boldsymbol{x}_i) & \mathbf{\Sigma}^{(h-1)}(\boldsymbol{x}_i, \boldsymbol{x}_j) \\ \mathbf{\Sigma}^{(h-1)}(\boldsymbol{x}_i, \boldsymbol{x}_j) & \mathbf{\Sigma}^{(h-1)}(\boldsymbol{x}_j, \boldsymbol{x}_j) \end{bmatrix} \in \mathbb{R}^{2 \times 2}, \tag{9}$$

$$\mathbf{\Sigma}^{(h)}(\boldsymbol{x}_i, \boldsymbol{x}_j) = \sigma_b^2 + \sigma_\omega^2 \mathbb{E}_{(u,v) \sim \mathcal{N}(0, \mathbf{\Lambda}^{(h)})} [\varphi(u)\varphi(v)]. \tag{10}$$

In order to compute the neural tangent kernel, derivative covariance is defined as

$$\dot{\mathbf{\Sigma}}^{(h)}(\boldsymbol{x}_i, \boldsymbol{x}_j) = \sigma_\omega^2 \mathbb{E}_{(u,v) \sim \mathcal{N}(0, \mathbf{\Lambda}^{(h)})} [\dot{\varphi}(u)\dot{\varphi}(v)]. \tag{11}$$

Then, with $\mathbf{\Theta}^{(0)}(\boldsymbol{x}_i, \boldsymbol{x}_j) = \mathbf{\Sigma}^{(0)}(\boldsymbol{x}_i, \boldsymbol{x}_j)$, the neural tangent kernel at each layer $\mathbf{\Theta}^{(h)}$ can be computed as follows

$$\mathbf{\Theta}^{(h)}(\boldsymbol{x}_i, \boldsymbol{x}_j) = \mathbf{\Sigma}^{(h)}(\boldsymbol{x}_i, \boldsymbol{x}_j) + \mathbf{\Theta}^{(h-1)} \dot{\mathbf{\Sigma}}^{(h-1)}(\boldsymbol{x}_i, \boldsymbol{x}_j). \tag{12}$$

The convergence of $\mathbf{\Theta}^{(L)}(\boldsymbol{x}_i, \boldsymbol{x}_j)$ to $\boldsymbol{H}_{ij}^*$ is proved in Theorem 3.1 in Arora et al. (2019).

## 3.1 Neural varifold computation

In this section, we present the kernel representation of varifold on point clouds via neural tangent kernel. We first introduce the neural tangent kernel representation of popular neural networks on point clouds (Qi et al., 2017a; Arora et al., 2019) by computing the neural tangent kernel for position and Grassmannian components, individually.

Given the set of $\hat{n}$ point clouds $\mathcal{S} = \{s_1, s_2, \cdots, s_{\hat{n}}\}$, where each point cloud $s_i = \{p_1, p_2, \cdots, p_{\hat{m}}\}$ is a set of points, and $\hat{n}, \hat{m}$ are respectively the number of point clouds and the number of points in each point cloud. Note that the number of points in each point cloud need not be the same (e.g., $|s_1| \neq |s_2|$). For simplicity, we assume below that different point clouds have the same number of points. Consider a PointNet-like architecture that consists of a $L$-hidden layers fully connected neural network shared by all points. For $(\hat{i}, \hat{j}) \in [\hat{m}] \times [\hat{m}]$, the covariance matrix $\mathbf{\Sigma}^{(h)}(p_{\hat{i}}, p_{\hat{j}})$ and neural tangent kernel $\mathbf{\Theta}^{(h)}(p_{\hat{i}}, p_{\hat{j}})$ at layer $h$ are defined and computed as in equation 10 and equation 12. Assuming that each point $p_{\hat{i}}$ consists of positional information and surface normal direction, i.e. $p_{\hat{i}} \in \mathbb{R}^3 \times \mathbb{S}^2$, the varifold representation can be defined with neural tangent kernel theory in two different ways. One way is to follow the Charon-Trouvé approach (Charon & Trouvé, 2013) by computing the position and Grassmannian kernels separately. While the original Charon-Trouvé approach uses the radial basis kernel for the positional elements and a Cauchy-Binet kernel for the Grassmannian parts, in our cases, we use the neural tangent kernel representation for both the positional and Grassmannian parts. Let $p_{\hat{i}} = \{x_{\hat{i}}, z_{\hat{i}}\} \in \mathbb{R}^3 \times \mathbb{S}^2$ be a pair of position $x_{\hat{i}} \in \mathbb{R}^3$ and its surface normal $z_{\hat{i}} \in \mathbb{S}^2$, $\hat{i} = 1, \ldots, \hat{m}$. The neural varifold representation is defined as

$$\mathbf{\Theta}^{\mathrm{varifold}}(p_{\hat{i}}, p_{\hat{j}}) = \mathbf{\Theta}^{\mathrm{pos}}(x_{\hat{i}}, x_{\hat{j}}) \cdot \mathbf{\Theta}^G(z_{\hat{i}}, z_{\hat{j}}). \tag{13}$$

We call the above representation PointNet-NTK1. As shown in Corollary 3.3 below, PointNet-NTK1 is a valid Charon-Trouvé type kernel. From the point of view of neural tangent theory, PointNet-NTK1 in equation 13 has two infinite-width neural networks on positional and Grassmannian components separately, and then aggregates information from the neural networks by element-wise product of the two neural tangent kernels.

**Corollary 3.3.** *In the limit of resolution going to infinity, the neural tangent kernels $\mathbf{\Theta}^{\mathrm{pos}}$ and $\mathbf{\Theta}^G$ are continuous positive definite kernels on positions and tangent planes, respectively. The varifold kernel $\mathbf{\Theta}^{\mathrm{varifold}} = \mathbf{\Theta}^{\mathrm{pos}} \odot \mathbf{\Theta}^G$ is a positive definite kernel on $\mathbb{R}^n \times \tilde{G}(d, n)$ and the associated RKHS $W$ is continuously embedded into $C_0(\mathbb{R}^n \times \tilde{G}(d, n))$.*

The other way to define a varifold representation is by treating each point as a 6-dimensional feature, with abuse of notation, $p_{\hat{i}} = \{x_{\hat{i}}, z_{\hat{i}}\} \in \mathbb{R}^6$. In this case, a single neural tangent kernel corresponding to an infinite-width neural network can be used, i.e.,

$$\mathbf{\Theta}^{\mathrm{varifold}}(p_{\hat{i}}, p_{\hat{j}}) = \mathbf{\Theta}(\{x_{\hat{i}}, z_{\hat{i}}\}, \{x_{\hat{j}}, z_{\hat{j}}\}). \tag{14}$$

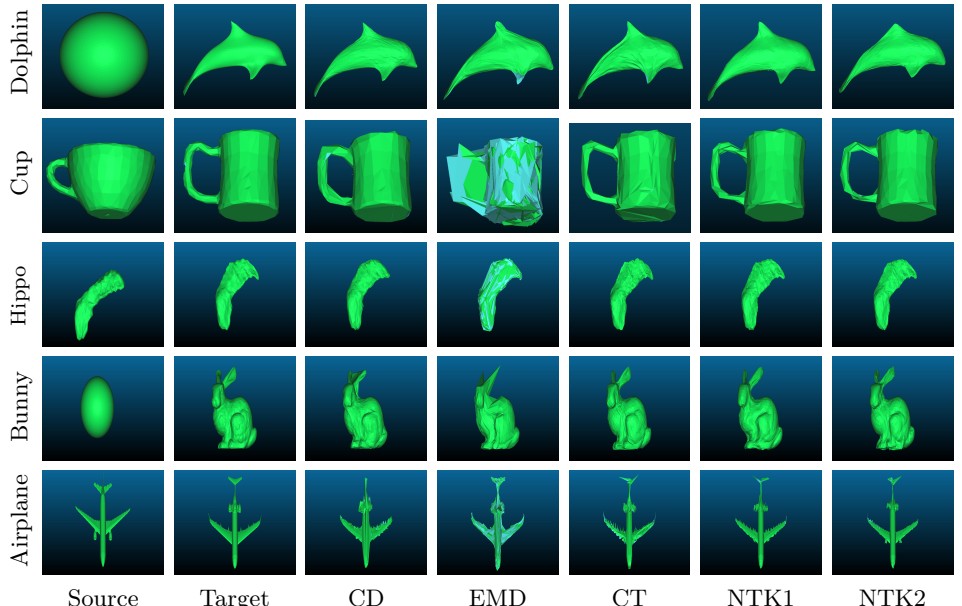

Figure 1: Shape matching examples with different shape similarity metrics, i.e., CD, EMD, CT, NTK1 and NTK2. Hippo is a shortened term referring to the hippocampus.

We call it PointNet-NTK2. Since PointNet-NTK2 does not compute the positional and Grassmannian kernels separately, it is computationally cheaper than PointNet-NTK1. It cannot be associated in the limit with a Charon-Trouvé type kernel, in contrast with PointNet-NTK1, but it remains theoretically well grounded because the explicit coupling of positions and normals is a key aspect of the theory of varifolds that provides strong theoretical guarantees (e.g., convergence, compactness, weak regularity, second-order information, etc.). Furthermore, PointNet-NTK2 falls into the category of neural networks proposed for point clouds (Qi et al., 2017a;b) that treat point positions and surface normals as 6-feature vectors, and thus PointNet-NTK2 is a natural extension of current neural networks practices for point clouds.

PointNet-NTK1 and PointNet-NTK2 in equation 13 and equation 14 are computing NTK values between two points $p_i$ and $p_{\hat{j}}$. The above forms can compute only pointwise-relationship in a single point cloud. However, in many point cloud applications, two or more point clouds need to be evaluated. Given the set of point clouds $\mathcal{S}$, one needs to compute a Gram matrix of size $\hat{n} \times \hat{n} \times \hat{m} \times \hat{m}$, which is computationally prohibitive in general. In order to reduce the size of the Gram matrix, we aggregate information by summation/average in all elements of $\boldsymbol{\Theta}^{\text{varifold}}$, thus forming an $\hat{n} \times \hat{n}$ matrix, i.e.,

$$\boldsymbol{\Theta}^{\text{varifold}}(s_i, s_j) = \sum_{\hat{i} \leq \hat{m}} \sum_{\hat{j} \leq \hat{m}} \boldsymbol{\Theta}^{\text{varifold}}(p_{\hat{i}} \in s_i, p_{\hat{j}} \in s_j). \tag{15}$$

Analogous to equation 3, the varifold representation $\boldsymbol{\Theta}^{\text{varifold}}$ can be used as a shape similarity metric between two sets of point clouds $s_i$ and $s_j$. The varifold metric can be computed as follows

$$\|s_i - s_j\|^2_{\text{varifold}} = \boldsymbol{\Theta}^{\text{varifold}}(s_i, s_i) - 2\boldsymbol{\Theta}^{\text{varifold}}(s_i, s_j) + \boldsymbol{\Theta}^{\text{varifold}}(s_j, s_j). \tag{16}$$

Furthermore, the varifold representation can be used for shape classification or any regression with the labels on point clouds data. Given training and test point cloud sets and their label pairs $(\boldsymbol{\chi}_{\text{train}}, \boldsymbol{\mathcal{Y}}_{\text{train}}) = \{(s_1, y_1), \cdots, (s_l, y_l)\}$ and $(\boldsymbol{\chi}_{\text{test}}, \boldsymbol{\mathcal{Y}}_{\text{test}}) = \{(s_{l+1}, y_{l+1}), \cdots, (s_{\hat{n}}, y_{\hat{n}})\}$, then neural varifold and its norm can be reformulated to predict labels using kernel ridge regression, i.e.,

$$\boldsymbol{\mathcal{Y}}_{\text{test}} = \boldsymbol{\Theta}^{\text{varifold}}_{\text{test}}(\boldsymbol{\chi}_{\text{test}}, \boldsymbol{\chi}_{\text{train}})(\boldsymbol{\Theta}^{\text{varifold}}_{\text{train}}(\boldsymbol{\chi}_{\text{train}}, \boldsymbol{\chi}_{\text{train}}) + \lambda \boldsymbol{I})^{-1} \boldsymbol{\mathcal{Y}}_{\text{train}}, \tag{17}$$

where $\lambda$ is the regularisation parameter.

# 4 Experiments

**Dataset and experimental setting.** We evaluate the varifold kernel representations and conduct comparisons on three different sought-after tasks: point cloud based shape matching between two different 3D meshes, point cloud based few-shot shape classification, and point cloud based 3D shape reconstruction. The details of each experiment setup are available in Appendix A.1, and the high-level pseudo-codes for each task are available at Appendex A.4. The ablation analysis of neural network depth, width, and hyperparameter settings is available in Appendix A.5. For ease of reference, we below shorten PointNet-NTK1, PointNet-NTK2, Chamfer distance, Charon-Trouvé varifold norm and Earth Mover's distance as NTK1, NTK2, CD, CT and EMD, respectively.

## 4.1 Shape matching

To evaluate the surface representation using neural varifolds and make comparison with existing shape similarity metrics, synthetic shape matching experiments are conducted. We train MLP networks with 2 hidden layers with width of 64 and 128 units respectively. These networks use various shape similarity metrics as loss functions to deform the given source shape into the target shape (more details are available in Appendix A.1.1).

Table 1: Results of shape matching deforming the given source shapes into the target shapes using a neural network trained with various shape similarity metrics. Metrics used in columns and rows are to train the neural network and for quantitative evaluation, respectively. Every value indicates the shape matching distance. In particular, the lowest and second lowest values (i.e., the best and the second best) in each row are highlighted in bold and underscored, respectively. Note that here "NTK1 and NTK2" are used as metrics.

|  | **Metric** | CD | EMD | CT | NTK1 | NTK2 |
|---|---|---|---|---|---|---|
| Dolphin | CD | **2.49E-4** | 3.39E-4 | 2.90E-4 | 2.84E-4 | 3.04E-4 |
|  | EMD | 7.56E0 | **3.87E0** | 4.15E0 | 4.13E0 | 4.27E0 |
|  | CT | 3.76E-2 | 2.94E-2 | **1.22E-2** | 1.63E-2 | 1.95E-2 |
|  | NTK1 | 6.56E-3 | 1.89E-3 | 2.93E-3 | **4.82E-4** | 6.34E-4 |
|  | NTK2 | 1.72E-2 | 4.33E-3 | 9.99E-3 | 1.34E-3 | **1.25E-3** |
| Cup | CD | 4.55E-3 | 9.74E-3 | 4.13E-3 | **3.26E-3** | 3.36E-3 |
|  | EMD | 2.03E1 | 3.53E1 | 2.06E1 | 1.85E1 | **1.79E1** |
|  | CT | 6.90E-1 | 2.85E0 | 4.07E-1 | 3.29E-1 | **3.20E-1** |
|  | NTK1 | 1.72E-2 | 7.27E-1 | 1.97E-2 | **6.07E-3** | 6.50E-3 |
|  | NTK2 | 3.14E-2 | 3.29E0 | 4.53E-2 | 1.34E-2 | **1.21E-2** |
| Hippocampus | CD | 3.49E-1 | 3.2E-1 | **2.43E-1** | 2.67E-1 | 2.65-1 |
|  | EMD | 2.80E5 | 2.10E5 | 2.25E5 | 2.09E5 | **1.96E5** |
|  | CT | 2.27E3 | 2.92E5 | 2.32E3 | 2.19E3 | **2.15E3** |
|  | NTK1 | 1.84E5 | 1.01E9 | 59.7E5 | **4.93E3** | 9.98E3 |
|  | NTK2 | 6.37E4 | 3.09E9 | 1.56E6 | **1.54E3** | **1.54E3** |
| Bunny | CD | 9.32E-3 | 5.12E-3 | **3.60E-3** | 4.40E-3 | 4.32E-3 |
|  | EMD | 2.31E4 | 4.74E3 | 3.72E3 | **3.13E3** | 3.52E3 |
|  | CT | 2.40E-1 | 1.25E0 | **7.51E-2** | 1.28E-1 | 1.23E-1 |
|  | NTK1 | 2.57E-2 | 1.32E-2 | 1.83E-3 | **2.22E-4** | 2.94E-4 |
|  | NTK2 | 3.85E-2 | 2.68E-2 | 3.33E-3 | 8.85E-4 | **6.43E-4** |
| Airplane | CD | 1.36E-3 | 4.07E-4 | **3.72 E-3** | 3.81E-3 | 5.90E-3 |
|  | EMD | 1.16E4 | 4.12E2 | **3.38E2** | 3.43E2 | 7.50E2 |
|  | CT | 8.71E-2 | 3.62E0 | **-3.58E-4** | 1.67E-3 | 3.68E-3 |
|  | NTK1 | 2.27E-3 | 1.80E-1 | 0.41E-6 | **0.31E-6** | 0.72E-6 |
|  | NTK2 | 6.14E-2 | 5.13E0 | 8.69E-6 | 3.17E-6 | **2.42E-6** |

Figure 1 shows five examples of shape matching based on various shape similarity metric losses. The neural network trained with CD captures geometric details well, except for the airplane. For hippocampi, CD oversmooths sharp edges; and for the bunny, it oversmooths the ears. While CD matches airplane wing shapes, it is noisier than CT, NTK1, and NTK2 methods. The EMD-trained network performs well on the dolphin shape but struggles with geometric details and surface consistency for other shapes, likely due to insufficient parameters for the transportation plan. More iterations and a lower convergence threshold make training inefficient. Networks trained with NTK1 and NTK2 metrics penalise broken meshes and surface noise, resulting in better mesh quality. NTK2 oversmooths high-frequency features on the dolphin, while NTK1 achieves good results. NTK1 and NTK2 show superior shape matching for airplane fuselage and wings. The network trained with CT gives acceptable results except for the airplane; however, one main disadvantage is that CT's radial basis kernel is sensitive to point cloud density, requiring hyperparameter $\sigma$ adjustments for each pair of point clouds to avoid poor results.

Table 1 presents the quantitative evaluation of the shape matching task. Each column indicates that the shape matching neural network is trained with a specific shape similarity metric as the loss function. In the case of dolphin, when the evaluation metric is the same as the loss function used to train the network, the network trained with the same evaluation metric achieves the best results. This is natural as the neural network is trained to minimise the loss function. It is worth highlighting that the shape matching network trained with

the NTK1 loss achieves the second best score for all evaluation metrics except for itself. In other words, NTK1 can capture common characteristics of all shape similarity metrics used to train the network. Furthermore, in the case of shape matching between two different cups, our neural varifold metrics (NTK1 and NTK2) achieve either the best or second best results regardless which shape evaluation metric is used. This indicates that the neural varifold metrics can capture better geometric details as well as surface smoothness for the cup shape than other metrics. In the case of shape matching between the source hippocampus and the target hippocampus, the network trained with CT excels in the CD metric, while the network trained with NTK1 achieves superior results with respect to NTK1 and NTK2 metrics. The shape matching network trained with NTK2 outperforms in the EMD, CT and NTK2 metrics. In the case of the bunny, CT shows the best results with respect to CD and CT, while NTK1 shows the best matching results with respect to EMD and NTK1. NTK2, on the other hand, shows the second best results with respect to all metrics except for itself. In the case of airplane, CT shows the best matching results with respect to CD, EMD and CT. However, the CT metric itself shows the negative value, i.e. unstable. This is mainly because the RBF kernel used in CT is badly scaled. NTK1 shows the second best shape matching results with respect to all metrics except for itself. The detailed analysis for the role of the NTK layers on shape matching is available in Appendix A.5.2.

## 4.2 Few-shot shape classification

In this section, the proposed NTKs are firstly compared with the current state-of-the-art few-shot classification methods on the ModelNet40-FS benchmark (Ye et al., 2023). ModelNet40-FS benchmark (Ye et al., 2023) divided different shape categories in ModelNet40 datasets for pre-training the network with 30 classes and then evaluated few-shot shape classification on 10 classes. The experiment was conducted in the standard few-shot learning setup, i.e. N-way K-shot Q-query. The definition of N-way K-shot Q-query is available in Appendix A.1.

Table 2 shows the shape classification results on two different few-shot classification setups, i.e., 5way-1shot-15query and 5way-5shot-15query. In the case of the 5way-1shot classification, the current state-of-the-art method PCIA achieves the best results by around 7% margin in comparison to the second best method NTK2 (pre-trained). In the case of the 5way-5shot classification, NTK2 outperforms PCIA by around 0.8% margin. Note

Table 2: Few-shot shape classification comparison on the ModelNet40-FS classification benchmark in terms of two setups, i.e., 5way-1shot and 5way-5shot. Every value indicates the mean shape classification accuracy with 95% confidence interval. NTK1 (DGCNN) and NTK2 (DGCNN) imply that, instead of point clouds positions and their normals, pointwise features from the pre-trained DGCNN are used for our NTK1 and NTK2.

| | ModelNet40-FS | |
|---|---|---|
| Methods | 5way-1shot | 5way-5shot |
| Prototypical Net | 69.96 ± 0.67 | 85.51 ± 0.52 |
| Relation Net | 68.57 ± 0.73 | 82.01 ± 0.53 |
| PointBERT | 69.41 ± 3.16 | 86.83 ± 2.03 |
| PCIA* | **82.21 ± 0.76** | 89.42 ± 0.53 |
| NTK1 | 64.94 ± 0.84 | 83.42 ± 0.59 |
| NTK2 | 62.67 ± 0.81 | 81.53 ± 0.59 |
| NTK1 (DGCNN) | 69.30 ± 0.76 | 86.75 ± 0.51 |
| NTK2 (DGCNN) | 75.23 ± 0.71 | **90.20 ± 0.49** |

\* Point cloud inputs are positions and unit normal vectors, i.e., 6-feature vectors. Note that the original paper's reported accuracy for 5way-1shot and 5way-5shot is 81.19% and 89.30%, respectively.

that PCIA requires to train backbone networks with PCIA modules and needs to fix the size of query. NTKs, on the other hand, can directly use the extracted backbone network features without further training the few-shot layer weights and do meta-learning in any arbitrary N-way K-shot Q-query settings. If NTKs are used without pre-trained backbone features, i.e., directly using positional and normal coordinates, then the results are subpar in comparison to other meta-learning approaches. This is understandable as few-shot architectures built on top of the backbone features, while NTKs without a pre-trained model, can only access the raw features, and thus cannot take advantages of the powerful feature learning capability of the neural networks. Interestingly, NTK1 outperforms NTK2 without pre-trained features, while NTK2 (DGCNN) outperforms NTK1 (DGCNN). This is because we use the pre-trained DGCNN on point clouds with spatial coordinates (x,y,z) as a backbone network for extracting both positional and normal features. Relatively low performance on NTK1 (DGCNN) is mainly because there is no appropriate architecture treating position and normal features separately.

Small-data tasks are common when data is limited. In the shape classification experiment, we restrict data availability and assume no pre-trained models, requiring training with 1, 5, 10, or 50 samples. Table 3 shows

ModelNet classification accuracy with limited samples. Kernel-based approaches excel in small-data tasks. In particular, with only one sample, kernel methods outperform finite-width neural networks like PointNet and DGCNN on both ModelNet10 and ModelNet40, with NTK2 and NTK1 achieving the best results, respectively. Interestingly, the CT kernel performs as well as NTK1 and NTK2 on ModelNet10 but drops significantly on ModelNet40. Similar results occur with five samples: NTK1 and NTK2 achieve 81.3% and 81.7% on ModelNet10, while CT, PointNet, and DGCNN lag by 3.1%, 5.1%, and 5.9%, respectively. On ModelNet40, NTK1 outperforms all other methods more significantly than on ModelNet10. As the number of training samples increases, finite-width neural networks significantly improve their performance on both ModelNet10 and ModelNet40. With ten samples, NTK1 and NTK2 achieve around 86.1% accuracy, outperforming other methods on ModelNet10 by 2–3%, although DGCNN surpasses NTK and PointNet on ModelNet40. With 50 samples, PointNet and DGCNN outperform NTK approaches by about 1% on ModelNet10 and 3–5% on ModelNet40. NTK1 and NTK2 show similar performance on ModelNet10 (with 0.3% difference), while NTK1 slightly outperforms NTK2 on ModelNet40 by 0.6–1.6%. Notably, NTK1 and NTK2 consistently outperform the CT varifold kernel.

Kernel-based learning is known for its quadratic computational complexity. However, NTK1 and NTK2 are computationally competitive in both few-shot learning and limited data scenarios. For instance, training NTK1 and NTK2 on ModelNet10 with 5 samples takes 47 and 18 seconds, respectively, compared to 254 and 502 seconds for training PointNet and DGCNN for 250 epochs on a single 3090 GPU. The shape classification performance on the full ModelNet data is available in Appendix A.3. Ablation study regarding the criteria used to choose the number of layers and different layer width for NTKs is available in Appendix A.5.

Table 3: ModelNet classification with limited training samples selected randomly. Every value indicates the average classification accuracy with standard deviation from 20 times iterations.

| Methods | 1-sample | 5-sample | 10-sample | 50-sample |
|---|---|---|---|---|
| | ModelNet10 | | | |
| PointNet | 38.84 ± 6.41 | 76.57 ± 2.28 | 84.14 ± 1.43 | 91.42 ± 0.89 |
| DGCNN | 33.56 ± 4.60 | 75.81 ± 2.40 | 83.90 ± 1.70 | **91.54 ± 0.68** |
| CT | 59.06 ± 4.76 | 78.64 ± 2.90 | 83.35 ± 1.57 | 87.98 ± 0.79 |
| NTK1 | 59.49 ± 4.80 | 81.34 ± 2.78 | 86.07 ± 1.62 | 90.18 ± 0.93 |
| NTK2 | **59.64 ± 5.50** | **81.74 ± 3.15** | **86.12 ± 1.56** | 90.10 ± 0.73 |
| | ModelNet40 | | | |
| PointNet | 33.11 ± 3.28 | 63.30 ± 2.12 | 73.63 ± 1.06 | 85.43 ± 0.31 |
| DGCNN | 36.04 ± 3.22 | 67.49 ± 1.80 | **77.04 ± 0.81** | **88.17 ± 0.57** |
| CT | 37.71 ± 3.42 | 60.43 ± 1.51 | 67.13 ± 1.11 | 77.20 ± 0.54 |
| NTK1 | **44.03 ± 3.51** | **69.30 ± 1.48** | 75.81 ± 1.23 | 83.88 ± 0.53 |
| NTK2 | 42.85 ± 3.51 | 67.81 ± 1.47 | 74.62 ± 1.00 | 83.26 ± 0.42 |

## 4.3 Shape reconstruction

Shape reconstruction from point clouds is tested for NTK1, SIREN, neural splines, and NKSR. NTK2 is excluded as it is unsuitable for this task. Implementation details are in Appendix A.2. Reconstruction quality is evaluated with CD and EMD metrics. Figure 2 shows examples (e.g., airplane and cabinet) with 2048 points. NTK1 performs better in surface completion and smoothness. Additional visualizations are in Appendix A.6.

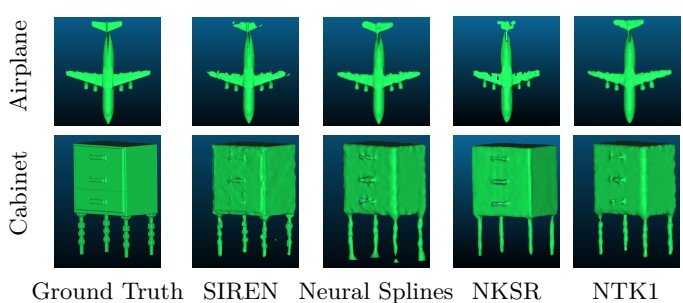

Ground Truth   SIREN   Neural Splines   NKSR   NTK1

Figure 2: Examples of the shape reconstruction comparison.

Quantitatively, Table 4 shows the mean and median of using the CD and EMD for 20 shapes randomly selected from each of the 13 different shape categories in the ShapeNet dataset. For the CD, NTK1 shows the best average reconstruction results for the airplane, cabinet, car and vessel categories; SIREN shows the best reconstruction results for the chair, display and phone categories; and the neural splines method shows the best reconstruction results for the rest 6 categories. NTK1 based reconstruction achieves the lowest mean EMD for vessel and cabinet, while

neural splines and SIREN achieve the lowest mean EMD for 7 and 5 categories, respectively. NKSR does not achieve the lowest mean CD and EMD for all the categories. In addition, the shape reconstruction results with different number of points (i.e., 512 and 1024) are available in Appendix A.5.5.

SIREN shows the lowest distance for both CD and EMD followed by NTK1. Surprisingly, the neural splines method underperforms in both the CD and EMD when we consider all the 13 categories. The performance of NTK1 on shape reconstruction is clearly comparable with these state-of-the-art methods. This might be counter-intuitive as it regularises the kernel with additional normal information, this is probably because there is no straightforward way to assign normals on the regular grid coordinates, where the signed distance values are estimated by the kernel regression.

Table 4: ShapeNet 3D mesh reconstruction with 2048 points (mean/median values ×1E3). NS: Neural Splines.

| Metric | Method | Airplane | Bench | Cabinet | Car | Chair | Display | Lamp | Speaker | Rifle | Sofa | Table | Phone | Vessel |
|---|---|---|---|---|---|---|---|---|---|---|---|---|---|---|
| CD (mean) | SIREN | 1.501 | 1.624 | 2.430 | 2.725 | **1.556** | **2.193** | 1.392 | 7.906 | 1.212 | 1.734 | 1.856 | **1.478** | 2.557 |
| | NS | 4.145 | **1.304** | 1.969 | 2.131 | 1.828 | 4.577 | **1.062** | **2.798** | **0.400** | **1.650** | **1.576** | 10.058 | 2.210 |
| | NKSR | 1.141 | 2.000 | 2.423 | 2.198 | 2.520 | 17.720 | 5.477 | 3.622 | 0.414 | 1.848 | 2.493 | 1.547 | 1.093 |
| | NTK1 | **0.644** | 1.314 | **1.991** | **2.107** | 1.734 | 4.666 | 1.134 | 2.806 | 0.425 | 1.654 | 1.586 | 10.397 | **1.079** |
| CD (median) | SIREN | 0.733 | 1.384 | 2.153 | 2.134 | 1.230 | 1.469 | 0.661 | 3.304 | 0.581 | 1.706 | 1.670 | 1.424 | 1.112 |
| | NS | 0.947 | 1.289 | 1.799 | **1.640** | **1.160** | 1.413 | **0.479** | **2.749** | 0.347 | 1.586 | 1.372 | 1.600 | **0.788** |
| | NKSR | 1.205 | 1.426 | **1.797** | 1.830 | 1.236 | 1.565 | 1.579 | 2.945 | **0.326** | 1.638 | 1.637 | **1.305** | 0.894 |
| | NTK1 | **0.621** | **1.259** | 1.828 | 1.836 | 1.237 | 1.499 | 0.566 | 2.794 | 0.352 | **1.578** | **1.350** | 1.558 | 0.797 |
| EMD (mean) | SIREN | **2.990** | 3.763 | 4.983 | 5.208 | **4.649** | **4.658** | 24.068 | 13.292 | 2.418 | **3.688** | 8.745 | **3.237** | 4.500 |
| | NS | 22.004 | **3.571** | **4.420** | **4.694** | 7.916 | 9.205 | **16.786** | **5.857** | 1.503 | 3.706 | **4.194** | 17.846 | 5.957 |
| | NKSR | 7.153 | 8.456 | 8.018 | 8.190 | 16.824 | 31.182 | 21.182 | 9.984 | 2.329 | 5.871 | 13.658 | 4.152 | 4.581 |
| | NTK1 | 3.120 | 4.153 | **4.420** | 4.767 | 7.350 | 9.653 | 23.381 | 6.236 | 1.592 | 3.888 | 5.259 | 24.101 | **3.534** |
| EMD (median) | SIREN | **2.690** | 2.938 | 4.520 | **3.803** | 4.411 | 3.314 | **2.279** | 6.240 | 1.605 | 3.653 | 3.782 | **3.060** | 2.576 |
| | NS | 6.873 | **3.068** | **4.154** | 3.999 | 4.740 | 4.053 | 3.802 | **5.123** | **1.216** | **3.543** | **3.695** | 3.838 | 2.210 |
| | NKSR | 5.732 | 5.119 | 4.440 | 5.313 | 5.683 | 3.777 | 4.927 | 5.975 | 1.227 | 3.641 | 6.375 | 3.088 | 2.771 |
| | NTK1 | 2.864 | 3.319 | 4.284 | 3.947 | 5.293 | 3.875 | 3.288 | 5.795 | 1.271 | 3.738 | 3.980 | 3.380 | **2.074** |

## 5 Limitation & Conclusion

The proposed method has few noticeable limitations. First, it is based on the simpler PointNet architecture. Future research should explore its performance with more advanced architectures like graph convolutions or voxelised point clouds. Second, the quadratic computational complexity of the kernel regime poses a challenge for large datasets. Kernel approximation methods, such as Nystrom approximation, could reduce this complexity, and their performance compared to exact kernels should be evaluated.

Despite the aforementioned limitations, the proposed method outperforms well-known approaches in the tasks of shape matching and few-shot shape classification, and shows competitive results in shape reconstruction. It is important to note that the tasks we evaluated represent only a small subset of the method's potential applications. For example, recent advances in Gaussian splatting explore the use of normal and depth information to enhance reconstruction quality (Turkulainen et al., 2024). The proposed neural varifold framework, which incorporates both spatial and normal information, can be applied in this context and extended to other modalities (e.g., depth, texture) with minor modifications (Hsieh & Charon, 2021).

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

# A Appendix

## A.1 Experimental setup

### A.1.1 Shape matching

For point cloud based shape matching, MLP networks consisting of 2 hidden layers (with width size of 64 and 128, respectively) were trained for computing displacement between two shapes, such that one can deform the source shape to the target shape. The neural networks were trained with different shape similarity metric losses including neural varifold. Point clouds of the given shapes were extracted by collecting the centre of triangular meshes of the given shapes, and the corresponding normals were computed by cross product of two edges of the meshes. The first example is deforming the source unit sphere into the target dolphin shape; the second is matching two different cup designs; the third is matching between two hippocampi; the fourth is the shape matching bewteen sphere and Stanford bunny; and the fifth is the shape matching between two different designs of airplane. The data is acquired from the PyTorch3D, SRNFmatch and KeOps GitHub repositories (Ravi et al., 2020; Martin Bauer & Hsieh, 2020; Charlier et al., 2021). This experiment evaluates how well the source shape can be deformed based on the chosen shape similarity measure as the loss function. A simple 3-layer MLP network was solely trained with a single shape similarity measure loss, with the learning rate fixed to 1E-3 and the Adam optimiser. The network was trained with popular shape similarity measures including the CD (Chamfer distance), EMD (Earth Mover's distance), CT (Charon-Trouvé varifold norm), and the proposed neural varifold norms (NTK1 and NTK2). In the case of CD and EMD, we followed the same method used for shape reconstruction. For varifold metrics, we used equation 16; note that it is a squared distance commonly used for optimisation. For the numerical evaluation as a metric in Table 1, the square-root of equation 16 was used. To be consistent with shape classification experiments, we chose the 5-layer NTK1 and 9-layer NTK2 to train and evaluate the similarity between two shapes. The detailed analysis for the role of the neural network layers on shape matching is available in Appendix A.5.2. The final outputs from the networks were evaluated with all of the shape similarity measures used in the experiments.

### A.1.2 Few-shot shape classification

The ModelNet40-FS dataset (Ye et al., 2023) was used in the case of evaluating few-shot learning capability of neural varifold kernels (NTK1 and NTK2) with popular few-shot learning methods including Prototypical Net (Snell et al., 2017), Relation Net (Sung et al., 2018), PointBERT (Yu et al., 2022), and PCIA (Ye et al., 2023). The ModelNet40-FS dataset (Ye et al., 2023) consists of 30 training and 10 unseen classes for training the backbone network and evaluating few-shot shape classification. The implementation of baseline methods and backbone networks is based on Ye et al. (2023). The computation of the neural varifold kernels (NTK1 and NTK2) is based on the *neural tangent* library (Novak et al., 2020). In this experiment, two different versions of NTK1 and NTK2s are used. First of all, NTK1 and NTK2 are directly computed from the original point cloud features (i.e., positions and their normals). As few-shot learning is usually based on pre-trained neural networks, NTK1 (DGCNN) and NTK2 (DGCNN) are computed from pointwise feature extracted from the backbone Dynamic graph convolutional neural network (DGCNN). DGCNN (Wang et al., 2019), used in the experiments, consists of 4 EdgeConv layers (Ye et al., 2023). Pointwise features are defined as the concatenation of the convolutional features extracted from all 4 EdgeConv layers of the DGCNN. Furthermore, global features are defined as the max-pooling of the pointwise features.

The evaluation was conducted using the standard few-shot classification setup: N-way K-shot Q-query. In this setup, N-way refers to the number of classes used for training and evaluation; K-shot indicates the number of samples per class used for training; and Q-query specifies the number of samples per class used for evaluating the classification accuracy. All methods are evaluated in two different few-shot learning scenarios: 5way-1shot-15query and 5way-5shot-15query. It is important to note that the reported accuracy in Table 2 represents the average accuracy and its 95% confidence intervals for 700 test cases (i.e., 700 test cases of N-way K-shot 15query).

In addition, we evaluate the scenario when pre-training data/models are not available. In this experimental setup, each method was also trained with a varying number of training samples per class, ranging from 1 to 50, and we evaluated their performance on the full ModelNet10/40 validation datasets. The number of

1024 points and their corresponding normals for each object were sampled from the original meshes of the Princeton ModelNet benchmark (Wu et al., 2015). The proposed neural varifold methods are compared with popular neural networks on point clouds including PointNet (Qi et al., 2017a), DGCNN (Wang et al., 2019), as well as the kernel method (Charon & Trouvé, 2013). The computation of the neural varifold kernels (NTK1 and NTK2) is based on the *neural tangent* library (Novak et al., 2020). To make the results more consistent, samples were randomly chosen and iterated 20 times with different seeds. Both NTK1 and NTK2 are required to fix the number of layers corresponding to the equivalent finite-width neural networks. NTK1 uses 5 fully connected neural network layers while NTK2 adopts 9 fully connected neural network layers. Each layer consists of MLP, layer normalisation and ReLU activation for both NTK1 and NTK2. The shape classification performance on the full ModelNet data is available in Appendix A.3. The criteria used to choose the number of layers and different layer width for both NTK1 and NTK2 are available in Appendix A.5.

### A.1.3 Shape reconstruction

Lastly, for shape reconstruction from point clouds, ShapeNet dataset (Chang et al., 2015) was used. In particular, we followed the data processing and shape reconstruction experiments from Williams et al. (2021), i.e., 20 objects from the individual 13 classes were randomly chosen and used for evaluating the shape reconstruction performance. For each shape, 2048 points were sampled from the surface and used for the reconstruction. Our approach was compared with the state-of-the-art shape reconstruction methods including Neural Splines (Williams et al., 2021), SIREN (Sitzmann et al., 2020) and neural kernel surface reconstruction (NKSR) (Huang et al., 2023). To be consistent with existing point cloud based shape reconstruction literature, CD and EMD were used to evaluate each method. Unlike CD, EMD has a number of different implementations for solving a sub-optimisation problem about the transportation of mass. In this study, we borrowed the EMD implementation code from Liu et al. (2020). In the experiment, we fixed the number of NTK1 network layers as 1. This is because there is no significant performance change when different number of network layers is used. The shape reconstruction using neural varifold is heavily influenced by the approaches from kernel interpolation (Cuomo et al., 2013) and neural splines (Williams et al., 2021). The implementation details are available in Appendix A.2. In addition, the shape reconstruction results with different number of points (i.e., 512 and 1024) are available in Appendix A.5.5. The visualisation of the ShapeNet reconstruction performance by all the methods compared is available in Appendix A.6.

## A.2 Kernel based shape reconstruction

Consider a set of surface points $\mathcal{X} = \{x_1, \cdots, x_k\}$ and their corresponding normals $\mathcal{Z} = \{z_1, \cdots, z_k\}$ sampled on an unknown surface $\mathcal{M}$, i.e., $\mathcal{X} \subset \mathcal{M}$. Using an implicit surface representation, all $x$ in $\mathcal{M}$ satisfy $f(x) = 0$ for some suitable function $f$. The best way to approximate the function $f$ is to generate off-surface points and to interpolate the zero iso-surface. Given $\mathcal{Y} = \{y_1, \cdots, y_k\}$, $\forall y_{\hat{i}} = 0$ and the distance parameter $\delta$, we define $\mathcal{X}_\delta^- = \{x_1 - \delta z_1, \cdots, x_k - \delta z_k\}$, $\mathcal{X}_\delta^+ = \{x_1 + \delta z_1, \cdots, x_k + \delta z_k\}$, $\mathcal{Y}_\delta^- = \{-\delta, \cdots, -\delta\}$, and $\mathcal{Y}_\delta^+ = \{\delta, \cdots, \delta\}$ in a similar manner. Taking the set unions $\hat{\mathcal{X}} = \mathcal{X} \cup \mathcal{X}_\delta^- \cup \mathcal{X}_\delta^+$, $\hat{\mathcal{Z}} = \mathcal{Z} \cup \mathcal{Z} \cup \mathcal{Z}$ and $\hat{\mathcal{Y}} = \mathcal{Y} \cup \mathcal{Y}_\delta^- \cup \mathcal{Y}_\delta^+$, the training data tuple $(\mathcal{X}_{\text{train}}, \mathcal{Y}_{\text{train}}) = (\{\hat{\mathcal{X}}, \hat{\mathcal{Z}}\}, \hat{\mathcal{Y}})$ (*cf.* symbols $\boldsymbol{\mathcal{X}}_{\text{train}}$ and $\boldsymbol{\mathcal{Y}}_{\text{test}}$ are used for multi-point clouds) can be used to obtain the *implicit representation of the surface.*

Let us define regular voxel grids $\mathcal{X}_{\text{grid}}$ on which all the extended point clouds $\hat{\mathcal{X}}$ lie. Note that there is no straightforward way to define normal vectors on the regular voxel grids, which are required for PointNet-NTK1 computation. Here, we assign their normals $\mathcal{Z}_{\text{grid}}$ as the unit normal vector to z-axis. Then the signed distance corresponding to the regular grid $\mathcal{X}_{\text{test}} = \{\mathcal{X}_{\text{grid}}, \mathcal{Z}_{\text{grid}}\}$ can be computed by kernel regression with neural splines or PointNet-NTK1 kernels $K(\mathcal{X}_{\text{train}}, \mathcal{X}_{\text{train}})$ and $K(\mathcal{X}_{\text{test}}, \mathcal{X}_{\text{train}})$, i.e.,

$$\mathcal{Y}_{\text{test}} = K(\mathcal{X}_{\text{test}}, \mathcal{X}_{\text{train}})[K(\mathcal{X}_{\text{train}}, \mathcal{X}_{\text{train}}) + \lambda \boldsymbol{I}]^{-1} \mathcal{Y}_{\text{train}}, \tag{18}$$

where $\mathcal{Y}_{\text{train}}$ and $\mathcal{Y}_{\text{test}}$ are the signed distances for the extended point clouds and the regular grids, respectively. With the marching cube algorithm in Lorensen & Cline (1998); Lewiner et al. (2003), the implicit signed distance values on the regular grid with any resolution can be reformulated to the mesh representation.

Table 5: ModelNet classification.

| Methods | ModelNet10 | ModelNet40 |
|---|---|---|
| PointNet*[1] | 94.4 | 90.5 |
| PointNet++[2] | 94.1 | 91.9 |
| DGCNN [3] | **95.0** | **92.2** |
| CT | 89.0 | 80.5 |
| **NTK1** | 92.2 | 87.4 |
| **NTK2** | 92.2 | 86.5 |

*Point cloud inputs are positions and unit normal vectors – 6-feature vectors; note that the original paper's reported accuracy for ModelNet40 is 89.2% with only positions forming 3-feature vectors as inputs.

## A.3 Shape classification with the full ModelNet dataset

The overall shape classification accuracy with neural varifold and the comparison with state-of-the-art methods on both ModelNet10 and ModelNet40 are given in Table 5, where the entire training data is used. The table shows that the finite-width neural network based shape classification methods (i.e., PointNet, PointNet++ and DGCNN) in general outperform the kernel based approaches, i.e., CT, NTK1 and NTK2. DGCNN shows the best accuracy on both ModelNet10 and ModelNet40 among the methods compared. In the case of kernel based methods, NTK1 outperforms both NTK2 and CT. The results are largely expected since the infinite-width neural networks with either NTK or NNGP kernel representations underperform in comparison with the equivalent finite-width neural networks (Lee et al., 2020) when sufficient training samples are available. The computational complexity of kernel-based approaches is quadratic. With the ModelNet10 dataset containing 4899 samples, NTK1 and NTK2 respectively require approximately 12 hours and 6 hours of training time, whereas PointNet and DGCNN achieve similar accuracy with nearly 1 hour of training time using the entire dataset.

Table 6: Shape classification performance of NTK1 and NTK2 with different number of neural network layers adopted in MLP and Conv1D on ModelNet40.

| Number of Layers | PointNet-NTK1 (5-sample) | PointNet-NTK2 (5-sample) |
|---|---|---|
| 1-layer MLP | $67.70 \pm 1.66$ | $64.70 \pm 1.34$ |
| 3-layer MLP | $69.06 \pm 1.57$ | $66.79 \pm 1.50$ |
| 5-layer MLP | $\mathbf{69.29 \pm 1.48}$ | $67.34 \pm 1.45$ |
| 7-layer MLP | $\mathbf{69.29 \pm 1.43}$ | $67.64 \pm 1.47$ |
| 9-layer MLP | $69.21 \pm 1.48$ | $\mathbf{67.81 \pm 1.47}$ |
| 1-layer Conv1D | $66.06 \pm 1.71$ | $63.20 \pm 1.30$ |
| 3-layer Conv1D | $68.82 \pm 1.62$ | $66.88 \pm 1.52$ |
| 5-layer Conv1D | $\mathbf{69.09 \pm 1.51}$ | $67.42 \pm 1.45$ |
| 7-layer Conv1D | $68.87 \pm 1.53$ | $67.77 \pm 1.41$ |
| 9-layer Conv1D | $68.68 \pm 1.46$ | $\mathbf{67.89 \pm 1.47}$ |

## A.4 Pseudo-code for PointNet-NTK Computation and Its Applications in Shape Matching, Classification, and Reconstruction

---

**Algorithm 1** PointNet-NTK Computations

---

**Require:** $s_i = \{\{x_1, z_1\}, \cdots, \{x_{\hat{m}}, z_{\hat{m}}\}\}, s_j = \{\{\hat{x}_1, \hat{z}_1\}, \cdots, \{\hat{x}_{\hat{m}}, \hat{z}_{\hat{m}}\}\}, N > 0$

  **if** PointNet-NTK1 **then**

    $\boldsymbol{X}, \hat{\boldsymbol{X}} \leftarrow \{x_1, x_2, \cdots, x_{\hat{m}}\}, \{\hat{x}_1, \hat{x}_2, \cdots, \hat{x}_{\hat{m}}\}$

    $\boldsymbol{Z}, \hat{\boldsymbol{Z}} \leftarrow \{z_1, z_2, \cdots, z_{\hat{m}}\}, \{\hat{z}_1, \hat{z}_2, \cdots, \hat{z}_{\hat{m}}\}$

    $\boldsymbol{\Theta}^{\text{pos}} \leftarrow$ **Algorithm 2** $(\boldsymbol{X}, \hat{\boldsymbol{X}}, N)$

    $\boldsymbol{\Theta}^{\text{nor}} \leftarrow$ **Algorithm 2** $(\boldsymbol{Z}, \hat{\boldsymbol{Z}}, N)$

    $\boldsymbol{\Theta}^{\text{varifold}} \leftarrow \boldsymbol{\Theta}^{\text{pos}} \odot \boldsymbol{\Theta}^{\text{nor}}$

  **else if** PointNet-NTK2 **then**

    $\boldsymbol{P} \leftarrow \{\text{CONCAT}(x_1, z_1), \cdots, \text{CONCAT}(x_{\hat{m}}, z_{\hat{m}})\}$

    $\hat{\boldsymbol{P}} \leftarrow \{\text{CONCAT}(\hat{x}_1, \hat{z}_1), \cdots, \text{CONCAT}(\hat{x}_{\hat{m}}, \hat{z}_{\hat{m}})\}$

    $\boldsymbol{\Theta}^{\text{varifold}} \leftarrow$ **Algorithm 2** $(\boldsymbol{P}, \hat{\boldsymbol{P}}, N)$

  **end if**

  **return** $\boldsymbol{\Theta}^{\text{varifold}}$

---

*Remark:* As an example, $\boldsymbol{X} \in \mathbb{R}^{\hat{m} \times 3}$ is formed by concatenating all $x_{\hat{i}} \in \{x_1, x_2, \cdots, x_{\hat{m}}\}$.

---

**Algorithm 2** NTK Corresponding to $N$-layer Infinite-width MLP with ReLU Activation$^*$

---

**Require:** $\boldsymbol{X}, \hat{\boldsymbol{X}}, N > 0$

  Initialise $\boldsymbol{\Theta}^{(0)} = \boldsymbol{\Sigma}^{(0)} = \boldsymbol{X}\hat{\boldsymbol{X}}^\top$, $\boldsymbol{d}_{\boldsymbol{X}}^{(0)} = (d_1^{(0)}, d_2^{(0)}, \cdots) = \text{diag}(\boldsymbol{X}\boldsymbol{X}^\top)$,

    $\hat{\boldsymbol{d}}_{\hat{\boldsymbol{X}}}^{(0)} = (\hat{d}_1^{(0)}, \hat{d}_2^{(0)}, \cdots) = \text{diag}(\hat{\boldsymbol{X}}\hat{\boldsymbol{X}}^\top)$

  **for** $h \leftarrow 1$ to $N$ **do**

    $\boldsymbol{\omega}_{\hat{i}, \hat{j}}^{(h-1)} \leftarrow \boldsymbol{\Sigma}_{\hat{i}, \hat{j}}^{(h-1)} / \sqrt{d_{\hat{i}}^{(h-1)} \hat{d}_{\hat{j}}^{(h-1)}}, \quad \hat{i}, \hat{j} = 1, 2, \ldots, \text{length}(\boldsymbol{d}_{\boldsymbol{X}}^{(h-1)})$

    $\dot{\boldsymbol{\Sigma}}^{(h-1)} \leftarrow \boldsymbol{F}_0(\boldsymbol{\Sigma}^{(h-1)}, \boldsymbol{d}_{\boldsymbol{X}}^{(h-1)}, \hat{\boldsymbol{d}}_{\hat{\boldsymbol{X}}}^{(h-1)})$, where $(\boldsymbol{F}_0)_{\hat{i}, \hat{j}} = 1 - \frac{1}{\pi} \arccos \boldsymbol{\omega}_{\hat{i}, \hat{j}}^{(h-1)}$

    $\boldsymbol{\Sigma}^{(h)} \leftarrow \boldsymbol{F}_1(\boldsymbol{\Sigma}^{(h-1)}, \boldsymbol{d}_{\boldsymbol{X}}^{(h-1)}, \hat{\boldsymbol{d}}_{\hat{\boldsymbol{X}}}^{(h-1)})$, where

        $(\boldsymbol{F}_1)_{\hat{i}, \hat{j}} = \frac{1}{2\pi} \sqrt{d_{\hat{i}}^{(h-1)} \hat{d}_{\hat{j}}^{(h-1)}} (\sqrt{1 - (\boldsymbol{\omega}_{\hat{i}, \hat{j}}^{(h-1)})^2} + (\pi - \arccos \boldsymbol{\omega}_{\hat{i}, \hat{j}}^{(h-1)}) \boldsymbol{\omega}_{\hat{i}, \hat{j}}^{(h-1)})$

    $\boldsymbol{\Theta}^{(h)} \leftarrow \boldsymbol{\Sigma}^{(h)} + \boldsymbol{\Theta}^{(h-1)} \dot{\boldsymbol{\Sigma}}^{(h-1)}$

    $\boldsymbol{d}_{\boldsymbol{X}}^{(h)}, \hat{\boldsymbol{d}}_{\hat{\boldsymbol{X}}}^{(h)} \leftarrow \frac{1}{2} \boldsymbol{d}_{\boldsymbol{X}}^{(h-1)}, \frac{1}{2} \hat{\boldsymbol{d}}_{\hat{\boldsymbol{X}}}^{(h-1)}$

  **end for**

  **return** $\boldsymbol{\Theta}^{(h)}$

---

$^*$Although **Algorithm 2** assumes the NTK representation corresponding to $N$-layer MLP with ReLU activation (Cho & Saul, 2009; Lee et al., 2019; Novak et al., 2020), several popular neural network layers have their corresponding closed-form NTK representations (Novak et al., 2020; Lee et al., 2020; Hron et al., 2020).

---

**Algorithm 3** Shape Matching

---

**Require:** $f(\cdot; \boldsymbol{\theta})$, $\mathcal{S}$, $\mathcal{T}$, $n_{\max} > 0$, $n_{\text{iter}} = 0$

  **while** $n_{\max} > n_{\text{iter}}$ **do**

    $v_{\mathcal{S}} \in \mathbb{R}^{|\mathcal{S}| \times 3}$ vertices of $\mathcal{S}$

    $d_{\mathcal{S}} \leftarrow f(v_{\mathcal{S}}; \boldsymbol{\theta})$ displacements between $\mathcal{S}$ and $\mathcal{T}$

    $\hat{\mathcal{S}} \leftarrow$ new source shape with deformed vertices $v_{\mathcal{S}} + d_{\mathcal{S}}$

    $\boldsymbol{x}_{\hat{\mathcal{S}}}, \boldsymbol{z}_{\hat{\mathcal{S}}} \leftarrow$ sample surface points and corresponding normals from $\hat{\mathcal{S}}$

    $\boldsymbol{x}_{\mathcal{T}}, \boldsymbol{z}_{\mathcal{T}} \leftarrow$ sample surface points and corresponding normals from $\mathcal{T}$

    $\hat{s}_{\hat{\mathcal{S}}} \leftarrow \{\boldsymbol{x}_{\hat{\mathcal{S}}}, \boldsymbol{z}_{\hat{\mathcal{S}}}\}$

    $\hat{s}_{\mathcal{T}} \leftarrow \{\boldsymbol{x}_{\mathcal{T}}, \boldsymbol{z}_{\mathcal{T}}\}$

    Compute $\|s_{\hat{\mathcal{S}}} - s_{\mathcal{T}}\|_{\text{varifold}}^2$ in equation 16 using **Algorithm 1**

    Backpropagate and update $\boldsymbol{\theta}$

    $\mathcal{S} \leftarrow \hat{\mathcal{S}}$

    $n_{\text{iter}} \leftarrow n_{\text{iter}} + 1$

  **end while**

---

*Remark:* Here $f(\cdot; \boldsymbol{\theta})$ is a 2-layer MLP neural network, $\boldsymbol{\theta}$ is the weights of the neural network $f$. $\mathcal{S}$ and $\mathcal{T}$ are the source and target shapes, respectively.

---

**Algorithm 4** Shape Classification

---

**Require:** $\boldsymbol{\mathcal{X}}_{\text{train}} = \{s_1, s_2, \cdots, s_l\}$, $\boldsymbol{\mathcal{Y}}_{\text{train}} = \{y_1, y_2, \cdots, y_l\}$, $\boldsymbol{\mathcal{X}}_{\text{test}} = \{s_{l+1}, s_{l+2}, \cdots, s_{\hat{n}}\}$, $N > 0$, where
$\quad s_i = \{p_1, p_2, \cdots, p_{\hat{m}}\}, i = 1, 2, \ldots, \hat{n}$

   **for** $i \leftarrow 1$ to $l$ **do**
      **for** $j \leftarrow 1$ to $l$ **do**
         $\boldsymbol{\Theta}^{\text{varifold}}(s_i, s_j) \leftarrow$ **Algorithm 1** $(s_i, s_j, N)$
         Aggregate points $\boldsymbol{\Theta}^{\text{varifold}}_{\text{train}_{(i,j)}} \leftarrow \sum_{\hat{i} \leq \hat{m}} \sum_{\hat{j} \leq \hat{m}} \boldsymbol{\Theta}^{\text{varifold}}(p_{\hat{i}} \in s_i, p_{\hat{j}} \in s_j)$
      **end for**
   **end for**
   **for** $i \leftarrow l + 1$ to $\hat{n}$ **do**
      **for** $j \leftarrow l + 1$ to $\hat{n}$ **do**
         $\boldsymbol{\Theta}^{\text{varifold}}(s_i, s_j) \leftarrow$ **Algorithm 1** $(s_i, s_j, N)$
         Aggregate points $\boldsymbol{\Theta}^{\text{varifold}}_{\text{test}_{(i,j)}} \leftarrow \sum_{\hat{i} \leq \hat{m}} \sum_{\hat{j} \leq \hat{m}} \boldsymbol{\Theta}^{\text{varifold}}(p_{\hat{i}} \in s_i, p_{\hat{j}} \in s_j)$
      **end for**
   **end for**
   $\boldsymbol{\mathcal{Y}}^{\text{pred}}_{\text{test}} \leftarrow \boldsymbol{\Theta}^{\text{varifold}}_{\text{test}}(\boldsymbol{\mathcal{X}}_{\text{test}}, \boldsymbol{\mathcal{X}}_{\text{train}})(\boldsymbol{\Theta}^{\text{varifold}}_{\text{train}}(\boldsymbol{\mathcal{X}}_{\text{train}}, \boldsymbol{\mathcal{X}}_{\text{train}}) + \lambda \boldsymbol{I})^{-1}\boldsymbol{\mathcal{Y}}_{\text{train}}$

---

**Algorithm 5** Shape Reconstruction[†]

---

**Require:** $\mathcal{X} = \{x_1, \cdots, x_k\}, \mathcal{Z} = \{z_1, \cdots, z_k\}, \mathcal{Y} = \{y_1, \cdots, y_k\}, \delta, \mathcal{X}_{\text{grid}}, \mathcal{Z}_{\text{grid}}, N > 0$
**Ensure:** $\forall y_{\hat{i}} = 0$ and $\delta > 0$
   $\mathcal{X}^-_\delta, \mathcal{X}^+_\delta \leftarrow \{x_1 - \delta z_1, \cdots, x_k - \delta z_k\}, \{x_1 + \delta z_1, \cdots, x_k + \delta z_k\}$
   $\mathcal{Y}^-_\delta, \mathcal{Y}^+_\delta \leftarrow \{-\delta, \cdots, -\delta\}, \{\delta, \cdots, \delta\}$
   $\hat{\mathcal{X}}, \hat{\mathcal{Z}}, \hat{\mathcal{Y}} \leftarrow \mathcal{X} \cup \mathcal{X}^-_\delta \cup \mathcal{X}^+_\delta, \mathcal{Z} \cup \mathcal{Z} \cup \mathcal{Z}, \mathcal{Y} \cup \mathcal{Y}^-_\delta \cup \mathcal{Y}^+_\delta$
   $\mathcal{X}_{\text{train}}, \mathcal{Y}_{\text{train}} \leftarrow \{\hat{\mathcal{X}}, \hat{\mathcal{Z}}\}, \hat{\mathcal{Y}}$
   $\boldsymbol{\Theta}^{\text{varifold}}(\mathcal{X}_{\text{train}}, \mathcal{X}_{\text{train}}) \leftarrow$ **Algorithm 1**$(\mathcal{X}_{\text{train}}, \mathcal{X}_{\text{train}}, N)$
   $\mathcal{X}_{\text{test}} \leftarrow \{\mathcal{X}_{\text{grid}}, \mathcal{Z}_{\text{grid}}\}$
   $\boldsymbol{\Theta}^{\text{varifold}}(\mathcal{X}_{\text{test}}, \mathcal{X}_{\text{train}}) \leftarrow$ **Algorithm 1** $(\mathcal{X}_{\text{test}}, \mathcal{X}_{\text{train}}, N)$
   $\mathcal{Y}^{\text{pred}}_{\text{test}} = \boldsymbol{\Theta}^{\text{varifold}}(\mathcal{X}_{\text{test}}, \mathcal{X}_{\text{train}})[\boldsymbol{\Theta}^{\text{varifold}}(\mathcal{X}_{\text{train}}, \mathcal{X}_{\text{train}}) + \lambda \boldsymbol{I}]^{-1}\mathcal{Y}_{\text{train}}$
   $\mathcal{S}_{\text{recon}} \leftarrow$ Marching cube algorithm (Lewiner et al., 2003) $(\mathcal{X}_{\text{test}}, \mathcal{Y}^{\text{pred}}_{\text{test}})$

[†]Please refer to a more detailed explanation of terms and equations in Appendix A.2.

---

### A.5 Ablation analysis

#### A.5.1 Neural varifolds with different number of neural network layers

This section shows the shape classification results based on different number of neural network layers. In this experiment, we randomly choose 5 samples per class on the training set of ModelNet40 and evaluate on its validation set. As shown in Section 4, we iterate the experiments 20 times with different random seeds. The key concept of the PointNet (Qi et al., 2017a) is the permutation invariant convolution operations on point clouds. For example, MLP or Conv1D with 1 width convolution window is permutation invariance. In this experiment, we choose different number of either MLP or Conv1D layers, and check how it performs on the ModelNet40 dataset. As shown in Table 6, the classification accuracy of NTK1 with Conv1D operation is lower in comparison with the ones with MLP layers. In particular, 5-layer and 7-layer MLPs show similar performance with the NTK1 architecture, i.e., 69.29% classification accuracy. In order to reduce the computational cost, we recommend fixing the number of layers in NTK1 to 5. In the case of NTK2, its performance increases as more layers are being added for it with both MLP and Conv1D operations. Furthermore, NTK2 with Conv1D operation shows slightly higher classification accuracy in comparison with the ones with MLP layers. The percentage of the performance improvement becomes lower as the number of layers increases. In particular, 9-layer MLP versus 7-layer MLP for NTK2 only brings 0.2% improvement; therefore, it is computationally inefficient to increase the number of layers anymore. Although NTK2 with 9-layer Conv1D achieves 0.08% higher accuracy than the one with 9-layer MLP, NTK2 with 9-layer MLP rather than Conv1D is used for the rest of the experiments in order to make the architecture consistent with the NTK1.

#### A.5.2 Shape matching with different number of neural network layers

Table 7: Ablation analysis for shape matching with respect to different number of neural network layers within NTK psueo-metrics. The number inside of the brackets (·) indicates the number of layers used for computing the NTK pseudo-metrics.

| Metric | NTK1 (1) | NTK1 (5) | NTK1 (9) |
|--------|----------|----------|----------|
| CD | 2.82E-1 | **2.67E-1** | 2.99E-1 |
| EMD | 2.43E5 | **2.09E5** | 2.46E5 |
| CT | 2.19E3 | **2.17E3** | **2.17E3** |
| NTK1 | 7.74E3 | 4.93E3 | **4.90E3** |
| NTK2 | 2.56E3 | **1.54E3** | 1.92E3 |

| Metric | NTK2 (1) | NTK2 (5) | NTK2 (9) |
|--------|----------|----------|----------|
| CD | **2.59E-1** | 2.61E-1 | 2.64E-1 |
| EMD | 2.14E5 | 2.32E5 | **1.93E5** |
| CT | **2.15E3** | 2.17E3 | **2.15E3** |
| NTK1 | 9.57E3 | **8.70E3** | 9.98E3 |
| NTK2 | **1.28E3** | 1.41E3 | 1.53E3 |

In this section, the behavior of the NTK pseudo-metrics with respect to different number of layers is evaluated. Note that the neural network width is not considered in this scenario as all pseudo-metrics are computed analytically (i.e., infinite-width). In this study, simple shape matching networks were trained solely by NTK psueo-metrics with different number of layers. Table 7 shows that the shape matching network trained with the 5-layer NTK1 metric achieves the best score with respect to CD, EMD, CT and NTK2 metrics, while the one with the 9-layer NTK1 metric achieves the best score with respect to CT and NTK1 metrics. This is in accordance with the ablation analysis for shape classification, where 5-layer NTK1 achieves the best classification accuracy in the ModelNet10 dataset. In comparison, NTK2 shows a mixed signal. The shape matching network trained with the 1-layer NTK2 metric achieves the best outcome with respect to Chamfer, CT and NTK2 metrics, while the one trained with the 9-layer NTK2 achieves the best results with respect to EMD and CT metrics. The network trained with 5-layer NTK2 shows the best result with respect to the NTK1 metric. This is not exactly in accordance with respect to shape classification with the NTK2 metric, where the shape classification accuracy improves as the number of layers increases. However, training a neural

network always involves some non-deterministic nature; therefore, it is yet difficult to conclude whether the number of neural network layers is important for improving the shape matching quality or not.

### A.5.3 Shape classification with hyperparameters $\sigma_w$ and $\sigma_b$

As discussed in the main Section 3 on the NTK, the computation depends on two hyperparameters representing the weight and bias variances in the infinite-width limit, denoted by $\sigma_w$ and $\sigma_b$, respectively. In this section, we evaluate the shape classification performance of a 9-layer MLP-based NTK2 by varying each hyperparameter independently. When varying $\sigma_w$, we fix $\sigma_b = 0.05$, and when varying $\sigma_b$, we fix $\sigma_w = 1$.

Table 8: Shape classification performance of 9-layer NTK2 with varying weight variance $\sigma_w$, fixed $\sigma_b = 0.05$.

| $\sigma_w$ | NTK2 Accuracy (5-sample) |
|---|---|
| 0.50 | $81.72 \pm 3.16$ |
| 1.00 | $81.73 \pm 3.16$ |
| 1.50 | $\mathbf{81.76} \pm 3.15$ |
| 2.00 | $81.75 \pm 3.16$ |
| 3.00 | $\mathbf{81.76} \pm 3.15$ |

Table 9: Shape classification performance of 9-layer NTK2 with varying bias variance $\sigma_b$, fixed $\sigma_w = 1$.

| $\sigma_b$ | NTK2 Accuracy (5-sample) |
|---|---|
| 0.00 | $\mathbf{81.76} \pm 3.15$ |
| 0.05 | $81.73 \pm 3.16$ |
| 0.10 | $81.72 \pm 3.16$ |
| 0.20 | $81.48 \pm 3.10$ |
| 0.50 | $77.59 \pm 3.01$ |

As shown in Tables 8 and 9, there is no significant variation in performance with changes in the hyperparameters. Although minor improvements are observed with different hyperparameter configurations, throughout all experiments in this paper, we adopt the standard parameterization ($\sigma_w = 1.0, \sigma_b = 0.05$) commonly used in the NTK literature (Lee et al., 2020; Novak et al., 2020). The theoretical interpretations of these NTK hyperparameters have been studied in the context of neural network generalizability and trainability (Xiao et al., 2020). From a computational perspective, the weight variance hyperparameter $\sigma_w$ acts as a multiplicative constant of 1.0 at each layer, while the bias variance $\sigma_b$ contributes by adding a constant term to all inputs.

### A.5.4 Shape classification with different neural network width

In this section, we analyse how the neural network width can impact on shape classification using the 9-layer MLP-based NTK2 by varying the width settings from 128, 512, 1024 and 2048 to infinite-width configurations. We trained the model on 5 randomly sampled point clouds per class from the ModelNet10 training set. The evaluation was carried out on the ModelNet10 validation set. This process was repeated five times with different random seeds, and the average shape classification accuracy was computed. Notably, NTK1 was excluded from this experiment due to the absence of a finite-width neural network layer corresponding to the elementwise product between two neural tangent kernels of infinite-width neural networks. The results presented in Table 10 demonstrate that the analytical NTK (infinite-width NTK) outperforms the empirical NTK computed from the corresponding finite-width neural network with a fixed width size. Furthermore, computing empirical NTK with respect to different length of parameters is known to be expensive as the empirical NTK is expressed as the outer product of the Jacobians of the output of the neural network with respect to the parameters. The details of the computational complexity and potential acceleration have been studied in Novak et al. (2022). However, if the finite-width neural networks are trained with the standard way instead of using empirical NTKs on a large dataset (e.g., CIFAR-10), then finite-wdith neural networks can outperform the neural tangent regime with performance significant margins (Lee et al., 2020; Arora et al., 2019). In other words, there is still a large gap understanding regarding training dynamics between the finite-width neural networks and their empirical neural kernel representations.

Table 10: Shape classification performance of 9-layer NTK2 with different neural network width.

| Width for each layer | NTK2 (5-sample) |
|---|---|
| 128-width | 78.74 ± 3.30 |
| 512-width | 80.08 ± 3.02 |
| 1024-width | 79.97 ± 3.24 |
| 2048-width | 80.46 ± 3.13 |
| infinite-width | **81.74 ± 3.16** |

Table 11: ShapeNet 3D mesh reconstruction with 1024 points (mean/median values ×1E3).

| Metric | Method | Airplane | Bench | Cabinet | Car | Chair | Display | Lamp | Speaker | Rifle | Sofa | Table | Phone | Vessel |
|---|---|---|---|---|---|---|---|---|---|---|---|---|---|---|
| CD (mean) | SIREN | **0.936** | **1.499** | 3.134 | 5.363 | **2.492** | **3.635** | 2.536 | 4.109 | 2.134 | 3.660 | 2.264 | **1.674** | 1.339 |
| | Neural Splines | 11.640 | 1.905 | **2.264** | 2.440 | 2.983 | 4.770 | **1.418** | **3.437** | **0.439** | 1.924 | 3.936 | 9.026 | 2.255 |
| | NKSR | 1.898 | 3.506 | 6.224 | **2.286** | 3.584 | 46.997 | 9.229 | 4.138 | 0.665 | 2.029 | 3.213 | 2.243 | **1.285** |
| | PointNet-NTK1 | 1.584 | 1.742 | 2.274 | 2.494 | 2.655 | 5.337 | 1.465 | 3.947 | 0.456 | **1.870** | **2.029** | 12.138 | 1.341 |
| CD (median) | SIREN | **0.756** | **1.272** | 2.466 | 2.305 | **1.281** | **1.385** | 1.156 | 3.411 | 0.487 | 1.706 | **1.601** | 1.390 | 1.040 |
| | Neural Splines | 8.171 | 1.562 | 1.830 | 2.058 | 2.152 | 1.548 | **0.698** | 3.071 | **0.359** | 1.657 | 1.715 | 1.594 | **0.879** |
| | NKSR | 1.900 | 2.245 | **1.799** | 2.190 | 2.116 | 1.880 | 2.347 | 3.488 | 0.407 | 1.697 | 1.695 | **1.345** | 0.956 |
| | PointNet-NTK1 | 0.820 | 1.701 | 1.933 | **1.995** | 1.522 | 1.719 | 0.733 | **3.045** | 0.366 | 1.719 | 1.643 | 1.658 | 1.016 |
| EMD (mean) | SIREN | **2.183** | **3.679** | 6.385 | 10.712 | **5.932** | **7.527** | **12.850** | 8.714 | 3.164 | 7.633 | **4.992** | **3.645** | **3.265** |
| | Neural Splines | 60.566 | 6.540 | 5.338 | **5.380** | 15.935 | 8.882 | 22.745 | **6.457** | 1.878 | **4.335** | 11.733 | 18.019 | 6.367 |
| | NKSR | 12.939 | 11.990 | 16.684 | 7.571 | 21.706 | 44.190 | 32.236 | 12.486 | 3.613 | 4.930 | 14.917 | 6.609 | 6.715 |
| | PointNet-NTK1 | 6.704 | 5.984 | **5.301** | 5.907 | 14.868 | 11.507 | 29.595 | 8.070 | **1.773** | 4.596 | 11.606 | 24.903 | 3.841 |
| EMD (median) | SIREN | **1.982** | 3.211 | 5.232 | 4.699 | **5.678** | **3.567** | **2.916** | 5.548 | 1.351 | 3.804 | **3.122** | 3.415 | 2.552 |
| | Neural Splines | 35.458 | 4.713 | **4.745** | 4.779 | 11.570 | 3.915 | 5.719 | **4.575** | **1.334** | 3.650 | 5.041 | 4.828 | 2.276 |
| | NKSR | 11.317 | 6.933 | 5.035 | 5.432 | 9.807 | 8.597 | 7.871 | 8.397 | 1.765 | **3.524** | 8.140 | **3.400** | 2.354 |
| | PointNet-NTK1 | 3.716 | 4.659 | 5.050 | **4.598** | 7.613 | 4.062 | 9.168 | 5.456 | 1.364 | 4.105 | 4.257 | 4.710 | **2.209** |

Table 12: ShapeNet 3D mesh reconstruction with 512 points (mean/median values ×1E3).

| Metric | Method | Airplane | Bench | Cabinet | Car | Chair | Display | Lamp | Speaker | Rifle | Sofa | Table | Phone | Vessel |
|---|---|---|---|---|---|---|---|---|---|---|---|---|---|---|
| CD (mean) | SIREN | **1.385** | **1.992** | 14.975 | 4.323 | **2.813** | **3.094** | 7.874 | 5.426 | 3.731 | 3.582 | 10.423 | **2.524** | 2.278 |
| | Neural Splines | 21.410 | 3.752 | 2.818 | 2.985 | 5.217 | 5.089 | **2.050** | 4.393 | 0.565 | **2.228** | 5.953 | 8.721 | 2.699 |
| | NKSR | 3.974 | 6.265 | 3.545 | **2.594** | 5.348 | NA | 9.859 | 5.259 | 17.419 | 2.059 | 6.636 | 1.677 | 1.540 |
| | PointNet-NTK1 | 2.454 | 2.674 | **2.565** | 3.233 | 3.793 | 6.087 | 2.193 | **4.045** | **0.550** | 2.252 | **2.702** | 14.349 | **2.090** |
| CD (median) | SIREN | **0.715** | **1.678** | 3.635 | 3.122 | **1.914** | **1.672** | 1.540 | 4.707 | 1.156 | 2.256 | **1.746** | 1.497 | 1.130 |
| | Neural Splines | 21.040 | 2.466 | 1.935 | 2.369 | 3.347 | 2.058 | **1.023** | 3.361 | **0.385** | 1.918 | 2.411 | 1.717 | 1.226 |
| | NKSR | 2.627 | 3.336 | **1.894** | 2.015 | 3.752 | NA | 4.427 | 3.753 | 0.906 | 1.833 | 3.555 | **1.411** | 0.856 |
| | PointNet-NTK1 | 1.243 | 2.246 | 2.106 | 2.316 | 2.473 | 1.968 | 1.346 | **3.330** | 0.387 | 1.890 | 1.963 | 2.013 | 1.309 |
| EMD (mean) | SIREN | **3.411** | **5.833** | 24.404 | 9.460 | **7.366** | **6.558** | **26.828** | 13.584 | 5.224 | 6.457 | 16.578 | 4.764 | **4.831** |
| | Neural Splines | 120.415 | 11.749 | 7.478 | **6.057** | 26.382 | 11.486 | 30.216 | 8.686 | 3.048 | **5.128** | 25.433 | 19.087 | 8.431 |
| | NKSR | 24.959 | 21.190 | 11.433 | 9.346 | 30.485 | NA | 36.050 | 18.147 | 13.115 | 5.226 | 24.257 | **4.701** | 8.605 |
| | PointNet-NTK1 | 13.826 | 9.217 | **5.614** | 11.548 | 16.465 | 13.501 | 35.540 | **8.334** | **2.436** | 6.010 | **15.663** | 27.025 | 5.897 |
| EMD (median) | SIREN | **1.964** | **5.036** | 8.656 | 6.643 | **5.553** | **3.650** | 14.281 | 14.499 | 2.296 | 4.682 | **3.735** | 3.779 | 3.012 |
| | Neural Splines | 115.527 | 9.698 | 4.679 | **4.863** | 20.006 | 4.476 | 10.834 | **5.405** | **1.548** | 4.234 | 8.205 | 4.742 | 3.147 |
| | NKSR | 25.234 | 14.795 | **4.405** | 6.669 | 16.082 | NA | 10.727 | 8.655 | 3.132 | **4.147** | 9.839 | **3.595** | **2.650** |
| | PointNet-NTK1 | 9.863 | 6.122 | 4.758 | 7.171 | 6.822 | 5.076 | **9.296** | 5.683 | 1.626 | 4.497 | 7.455 | 6.658 | 3.313 |

NA indicates that the method fails to reconstruct few shapes in the given class.

### A.5.5 Shape reconstruction with different point cloud sizes

In this section, we compare shape reconstruction results with different point cloud sizes, i.e., 512, 1024 and 2048 points. As indicated in Tables 4, 11 and 12, NTK1 and neural splines show that the quality of the reconstructions is degraded as the number of points decreases. For NKSR, its reconstruction quality becomes worse as the number of point clouds decreases for most categories, but few categories (i.e., cabinet and vessel) show the opposite trend. In the case of SIREN, the convergence of the SIREN network plays more important role for the shape reconstruction quality. For example, the shape reconstruction results by SIREN on the airplane category show that the shape reconstruction with 1024 points is better than that with 2048 points. This is due to the non-deterministic nature of DNN libraries, i.e., it is difficult to control the convergence of the SIREN network with our current experimental setting $10^4$ epochs. Note that the SIREN reconstruction

is computationally much more expensive (around 20∼30 minutes) than either the NTK1, neural splines or the NKSR approach (around 1∼5 seconds).

## A.6 Visualisation of ShapeNet reconstruction results

In this section, we present additional visualisations of shape reconstruction outcomes obtained through three baseline methods (i.e., SIREN, neural splines, and NKSR), along with the proposed NTK1 method, across 13 categories of ShapeNet benchmarks. Five shape reconstruction results are illustrated for each category. Specifically, Figure 3 showcases examples from the Airplane, Bench, and Cabinet categories. Figure 4 exhibits five instances of shape reconstruction outcomes for the Car, Chair, and Display categories. Moving on to Figure 5, it displays examples from the Lamp, Speaker, and Rifle categories. Similarly, Figure 6 demonstrates five instances of shape reconstruction results for the Sofa, Table, and Phone categories. Finally, Figure 7 focuses on the shape reconstruction results for the Vessel category.

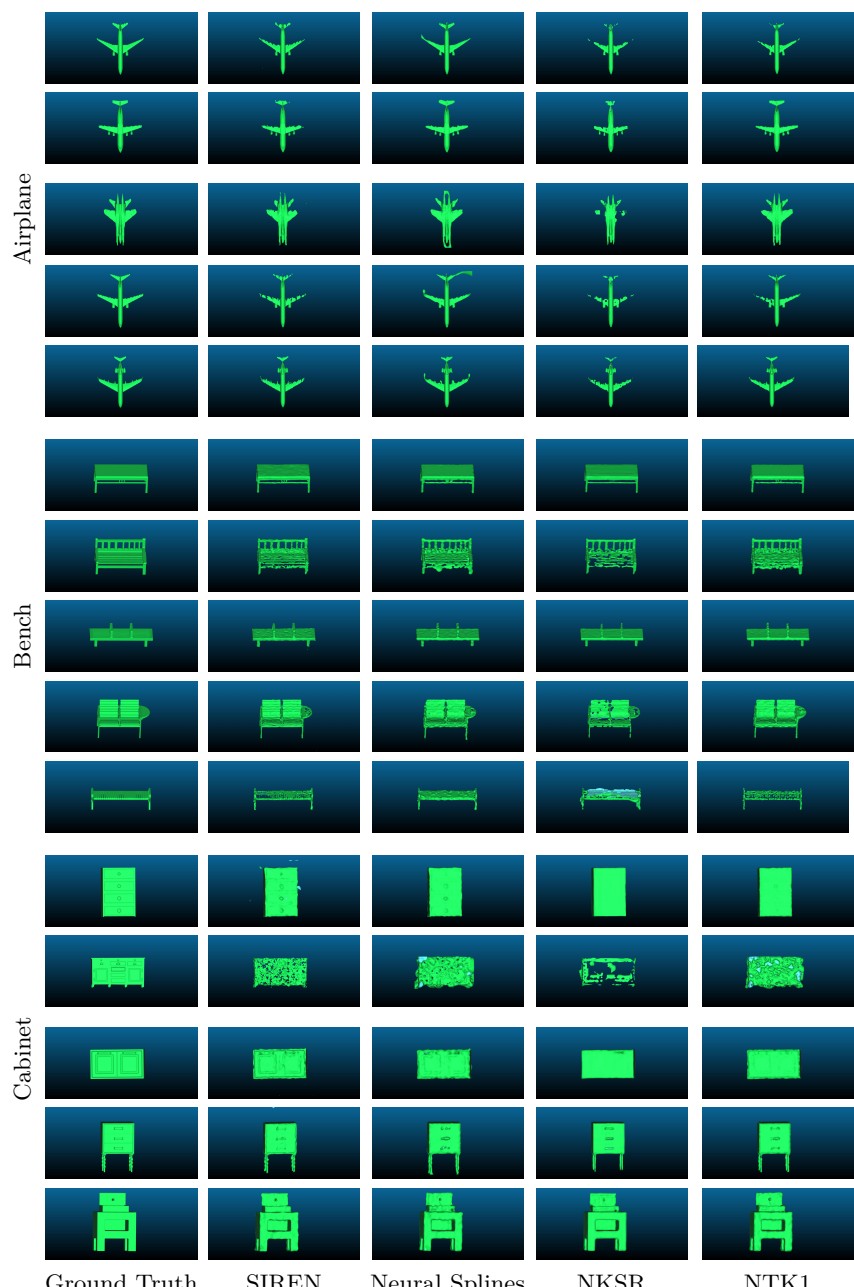

Figure 3: Visualisation of shape reconstruction results from SIREN, Neural Splines, NKSR and NTK1 for the Airplane, Bench and Cabinet categories.

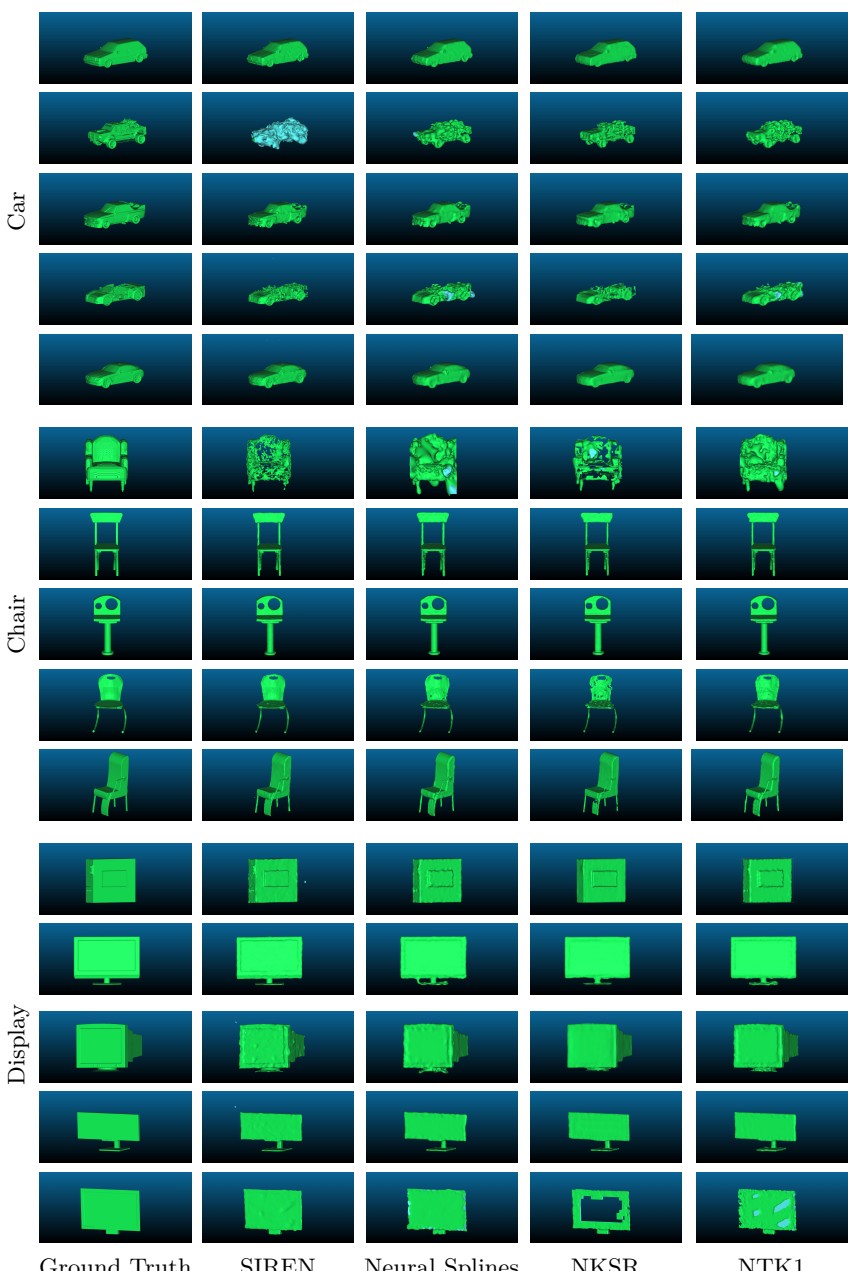

Figure 4: Visualisation of shape reconstruction results from SIREN, Neural Splines, NKSR and NTK1 for the Car, Chair and Display categories.

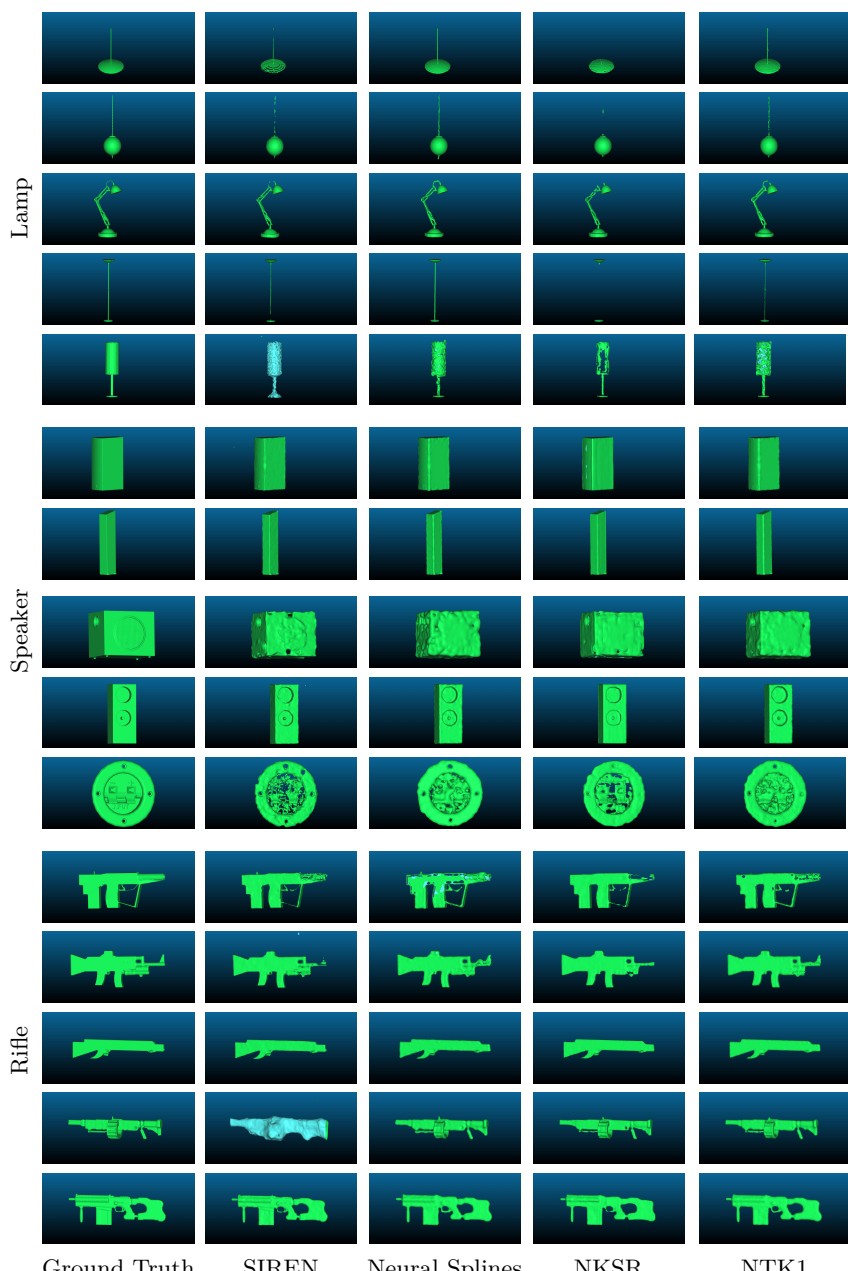

Figure 5: Visualisation of shape reconstruction results from SIREN, Neural Splines, NKSR and NTK1 for the Lamp, Speaker and Rifle categories.

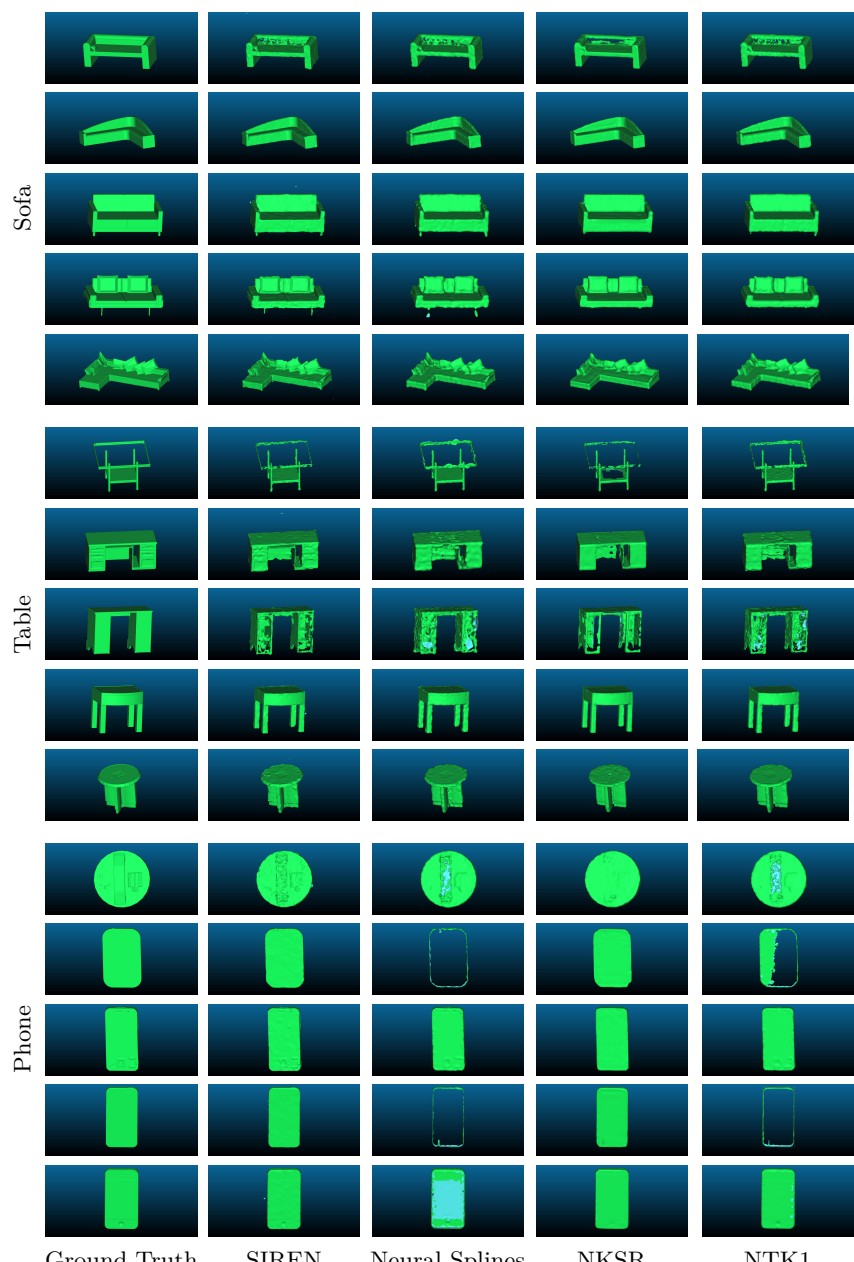

Figure 6: Visualisation of shape reconstruction results from SIREN, Neural Splines, NKSR and NTK1 for the Sofa, Table and Phone categories.

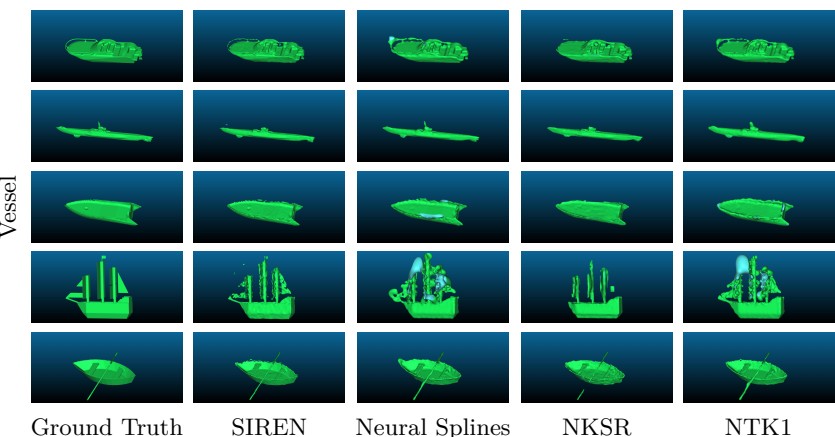

Figure 7: Visualisation of shape reconstruction results from SIREN, Neural Splines, NKSR and NTK1 for the Vessel category.

