# OpenReview forum: "Neural varifolds: an aggregate representation for quantifying the geometry of point clouds"
_TMLR — Accepted by TMLR_

### Review · Reviewer_QcnL · 2025-03-19

**Summary Of Contributions:**

This paper introduces a neural varifold representation of 3d point cloud. The authors propose neural varifold algorithms to compute the varifold norm between point clouds using neural networks and neural tangent kernel. Extensive experiments are conducted for the proposed neural varifold on three point cloud tasks to demonstrate the effectiveness of the proposed representations.

**Audience:**

Yes

**Broader Impact Concerns:**

No obvious concerns on the ethical implications of this work.

**Claims And Evidence:**

Yes

**Requested Changes:**

1. The $p_\hat{i}$ in equation (13) and equation (14) is different, but with same notation. It would be better to use different notation to distinguish them.
2. Equation (16) does not make sense. Might have typos with $s_i$ and $s_j$.
3. In section 4 - Dataset and experimental setting, it would be better if the authors could introduce more on shape similarity metrics, and highlight the difference with the proposed one. Could also introduce this part in the Related Work section.
4. In Table1, NTK1 and NTK2 are used as metrics, while in Table 2, they are used as methods. It would be better if the authors could distinguish them to avoid confusion.
5. It would be better if the authors to include a Conclusion section to highlight the contributions and takeaways.

**Strengths And Weaknesses:**

Strength:
1. The paper is well-written, with clear structure and illustration.
2. The experiments are thorough and comprehensive. Additional results, ablation studies and visualizations are provided.

Weakness:
1. The architecture assumption in section 3 (Neural tangent kernel, page 4) is pretty strong. It is not clear whether this representation could be extended to more advanced/complicated architectures.
2. Comparing to evaluation metrics, like CD and EMD, additional information (surface normal) is needed. In the real-world application, these information may not be available/expensive to acquire.
3. The related work on different representations/evaluation metrics are not throughly discussed.

---

> ### Author Response · Authors · 2025-04-26
>
> **Weakness. The architecture assumption in section 3 (Neural tangent kernel, page 4) is pretty strong. It is not clear whether this representation could be extended to more advanced/complicated architectures.**
>
>  The underlying assumption in neural tangent kernel (NTK) theory is indeed strong. Its application for the various architectures is an active area of the research. For example, it has been extended for general graph neural network architecture [1], attention algorithm [2], and recurrent neural network [3]. We would also like to put an emphasis on that NTK theory has been used to understand the learning dynamics of many popular real-world architectures (e.g. CNN [4], GAN [5], and LLM fine tuning [6]).
>
> In the revision on page 10, we have added Section 5 for "limitation and conclusion".
>
> **Weakness. Comparing to evaluation metrics, like CD and EMD, additional information (surface normal) is needed. In the real-world application, these information may not be available/expensive to acquire.**
>
> The reviewer is indeed correct that surface normal information is often not available or difficult to compute. However, we believe there are also various applications that take the normal information available, for example, meshes and CAD. Moreover, there are a reasonable number of literature investigating its potential applications on shape matching and geometric analysis [7,8]. In the case of point clouds, it is often harder to approximate unless the point clouds are dense enough. In modern applications, recently there exists literature exploring the use of additional normal information to improve 3D Gaussian Splatting technique or diffusion models [9,10,11], where our approach may provide some insights for the potential applications.
>
> **The $p_i$ in equation (13) and equation (14) is different, but with same notation. It would be better to use different notation to distinguish them.**
>
> In our revision, we have added the phrase "with abuse of notation" to avoid confusion. We feel it is better to use the same notation; otherwise, the notation will become unnecessarily complex for different cases.
>
> **Equation (16) does not make sense. Might have typos with  $s_i$ and $s_j$.**
>
> Thanks for the reviewer's comment. We corrected the index $i,j$ accordingly in Equation (16) in the revision.
>
> **In section 4 - Dataset and experimental setting, it would be better if the authors could introduce more on shape similarity metrics, and highlight the difference with the proposed one. Could also introduce this part in the Related Work section.**
>
> **Weakness. The related work on different representations/evaluation metrics are not throughly discussed.**
>
>  We appreciate the reviewer's point. We added another paragraph in the related work section on page 2 in the revision discussing the shape similarity.
>
> **In Table1, NTK1 and NTK2 are used as metrics, while in Table 2, they are used as methods. It would be better if the authors could distinguish them to avoid confusion.**
>
> Thanks for the reviewer's suggestion. In the revision, at the end of the caption of Table 1 on page 7, we highlighted that "Note that here "NTK1 and NTK2" are used as metrics" to avoid any confusion.
>
> **It would be better if the authors to include a Conclusion section to highlight the contributions and takeaways.**
>
> In the revision on page 10, we have added Section 5 for "limitation and conclusion".

---

> ### Author Response · Authors · 2025-04-26
>
> **references**
>
> [1] Du, Simon S., et al. "Graph neural tangent kernel: Fusing graph neural networks with graph kernels." Advances in Neural Information Processing Systems 32 (2019).
>
> [2] Hron, Jiri, et al. "Infinite attention: NNGP and NTK for deep attention networks." International Conference on Machine Learning. PMLR, 2020.
>
>
> [3] Alemohammad, Sina, et al. "Scalable neural tangent kernel of recurrent architectures." arXiv preprint arXiv:2012.04859 (2020).
>
> [4] Xiao, Lechao, Jeffrey Pennington, and Samuel Schoenholz. "Disentangling trainability and generalization in deep neural networks." International Conference on Machine Learning. PMLR, 2020.
>
> [5] Franceschi, Jean-Yves, et al. "A neural tangent kernel perspective of GANs." International Conference on Machine Learning. PMLR, 2022.
>
> [6] Ren, Yi, and Danica J. Sutherland. "Learning dynamics of llm finetuning." arXiv preprint arXiv:2407.10490 (2024).
>
> [7] Vaillant, Marc, and Joan Glaunes. "Surface matching via currents." Biennial International Conference on Information Processing in Medical Imaging. Berlin, Heidelberg: Springer Berlin Heidelberg, 2005.
>
> [8] Charon, Nicolas, and Alain Trouvé. "The varifold representation of nonoriented shapes for diffeomorphic registration." SIAM Journal on Imaging Sciences, 6.4 (2013): 2547-2580.
>
> [9] Turkulainen, Matias, et al. "Dn-splatter: Depth and normal priors for Gaussian splatting and meshing." arXiv preprint arXiv:2403.17822 (2024).
>
> [10] Huang, Xin, et al. "Humannorm: Learning normal diffusion model for high-quality and realistic 3d human generation." Proceedings of the IEEE/CVF Conference on Computer Vision and Pattern Recognition. 2024.
>
> [11] Wei, Meng, et al. "Normal-GS: 3D Gaussian Splatting with Normal-Involved Rendering." arXiv preprint arXiv:2410.20593 (2024).

---

### Review · Reviewer_qt59 · 2025-03-27

**Summary Of Contributions:**

The paper introduces a novel approach to quantifying the geometry of point clouds using neural varifolds, which combine deep learning techniques with geometric fidelity metrics. The key contributions are:

1. The authors propose a new representation for point clouds that captures both point positions and tangent spaces, leveraging the concept of varifolds from geometric measure theory.

2. Two algorithms are proposed to compute the varifold norm between two point clouds using neural networks and neural tangent kernel representations.

3. The neural varifold representation is evaluated on three tasks: shape matching, few-shot shape classification, and shape reconstruction. The results demonstrate that the proposed method outperforms state-of-the-art methods in shape matching and few-shot shape classification, and is competitive in shape reconstruction.

**Audience:**

Yes

**Claims And Evidence:**

Yes

**Requested Changes:**

1. The authors should evaluate the performance of kernel approximation methods (e.g., Nystrom approximation) to address the computational complexity issue. This would strengthen the paper by providing a practical solution for scaling the method to larger datasets. The time complexity should be fully evaluated.

2. The authors should explore the performance of the neural varifold representation with more advanced architectures, such as graph convolutions or voxelized point clouds. This would demonstrate the generalizability of the proposed method.

3. The authors should provide a more detailed analysis of the sensitivity of the proposed neural varifold methods to hyperparameters, particularly in comparison to the Charon-Trouve varifold norm.

**Strengths And Weaknesses:**

Strengths:**

1. The introduction of neural varifolds is a novel and promising approach to representing and analyzing point cloud data, particularly in capturing both macro and subtle geometric features. The paper is well-grounded in geometric measure theory, providing a solid theoretical foundation for the proposed methods.

2. The neural varifold representation is shown to be effective across multiple tasks, including shape matching, few-shot classification, and shape reconstruction, demonstrating its versatility.


**Weaknesses:**

1. The running time should be evaluated for the proposed neural varifold approach, which could be a limitation for large data. While kernel approximation methods are mentioned, their performance is not evaluated in the paper.

2. The proposed method is based on the simpler PointNet architecture. The paper suggests that future work could explore more advanced architectures like graph convolutions or voxelized point clouds, but this is not addressed in the current work. It is encouraged that to evaluate the more divese network architectures and release the code for community testing.

3. As a measurement of quantifying the geometry of point cloud, the parameter sensity is a very essential part. It is good if the neural varifold is robust for different cases. The paper mentions that the Charon-Trouve varifold norm is sensitive to point cloud density and requires hyperparameter adjustments for each pair of point clouds. It is unclear if the proposed neural varifold methods share similar sensitivity.

---

> ### Author Response · Authors · 2025-04-26
>
> **Weakness The running time should be evaluated for the proposed neural varifold approach, which could be a limitation for large data. While kernel approximation methods are mentioned, their performance is not evaluated in the paper**
>
> **The authors should evaluate the performance of kernel approximation methods (e.g., Nystrom approximation) to address the computational complexity issue. This would strengthen the paper by providing a practical solution for scaling the method to larger datasets. The time complexity should be fully evaluated**
>
>
> The quadratic complexity of kernel methods is indeed a significant problem for kernel based approaches in the context of their applications to large datasets. Furthermore, in the context of neural tangent kernel theory, it is well known that even with exact kernel computation, infinite-width neural networks generally do not achieve classification accuracy on large datasets as high as their finite-width counterparts [1,2]. For example, in CIFAR-10 classification, the standard LeNet architecture significantly outperforms its NTK computation by a large margin and the computational costs for the standard neural network are significantly cheaper [1,2]. Thus, our study was mainly focused on the tasks where the kernel approach can compete with the standard neural network approaches, e.g., shape similarity metrics, few-shot classification, and shape reconstruction.
>
> In the case of the few-shot classification (e.g., 1-shot, and 5-shot), kernel approximation might lose too much information. As we shown in the paragraph, its computation is in general affordable, and thus we have not looked into the kernel approximation in this study. However, we agree that it would be interesting to pursue in the direction for optimizing kernel approximation and make the algorithm more efficient for larger datasets.  In the context of the shape similarity measure, it would be very interesting to know how it accurately approximates the kernel pseudo-metric, but this requires more rigorous analysis. In the case of the shape reconstruction, as it works on the individual points rather than a set of several point clouds, parallelization and Nystrom approximation could be used and indeed make the proposed algorithm more practical. Currently our method is based on neural tangent library [3], which does not support matrix approximation methods, but it can be re-implemented to the FALKON [4] or KeOps [5], where more powerful GPU parallelization as well as kernel approximation methods are supported. As we are going to release our code upon the publication, we will support kernel based approximation as a follow-up.
>
> **Weakness. The proposed method is based on the simpler PointNet architecture. The paper suggests that future work could explore more advanced architectures like graph convolutions or voxelized point clouds, but this is not addressed in the current work. It is encouraged that to evaluate the more divese network architectures and release the code for community testing.**
>
> **The authors should explore the performance of the neural varifold representation with more advanced architectures, such as graph convolutions or voxelized point clouds. This would demonstrate the generalizability of the proposed method.**
>
>
> As the reviewer mentioned, the NTK form of more advanced architectures (e.g., graph neural network [6], and infinite-width attentions [7]) is available. However, unlike the PointNet architecture where its kernel computation is relatively simple, more advanced architectures require further computational optimization. Otherwise, their kernel form computation is in general very expensive. For example, graph neural network requires to compute the graph structure first and we need to store them, as well as propagates over the quadratic kernel representations. It requires additional complexity to the experiments. In particular, which graph neural network architecture we are going to choose, what the graph size is, and whether this is KNN graph or radius graph, etc.; there are many hyperparameters to control and optimize. In this study, we are proposing the neural varifold as a concept as well as a metric with relatively simpler hyper-parameterization; therefore, we choose relatively simple but efficient architecture (PointNet).

---

> ### Author Response · Authors · 2025-04-26
>
> **Weakness. As a measurement of quantifying the geometry of point cloud, the parameter sensity is a very essential part. It is good if the neural varifold is robust for different cases. The paper mentions that the Charon-Trouve varifold norm is sensitive to point cloud density and requires hyperparameter adjustments for each pair of point clouds. It is unclear if the proposed neural varifold methods share similar sensitivity.**
>
> **The authors should provide a more detailed analysis of the sensitivity of the proposed neural varifold methods to hyperparameters, particularly in comparison to the Charon-Trouve varifold norm**
>
> As the reviewer pointed out, the hyperparameter $\sigma$ is a critical factor in the performance of the radial basis function (RBF) kernel. Due to its exponential form, the RBF kernel value decays rapidly with increasing geometric distance between input points, making it highly sensitive to point cloud density and local geometry.
>
> In contrast, the NTK is generally dense, and its notion of locality is determined by the neural network architecture it corresponds to. For instance, in the case of the PointNet architecture, the NTK is derived from the infinite-width limit of the network, requiring appropriately scaled weight and bias parameters $w$ and $b$ at each layer. In our experiments, we adopted the standard parameterization ($w=1$ and $b=0.05$) from the NTK literature [1,3]. The effects of these NTK hyperparameters have been investigated in the context of generalizability and trainability of the neural networks [8].  In the sense of the computation, weight hyper-parameter terms are multiplying constant 1.0 at each layer, and bias term contributes to all input values by adding constant $0.05^2$.
>
>
> **references**
>
>
> [1] Lee, Jaehoon, et al. "Finite versus infinite neural networks: an empirical study." Advances in Neural Information Processing Systems 33 (2020): 15156-15172.
>
> [2] Shankar, Vaishaal, et al. "Neural kernels without tangents." International Conference on Machine Learning. PMLR, 2020.
>
>
> [3] Novak, Roman, et al. "Neural tangents: Fast and easy infinite neural networks in python." International Conference on Learning Representation (2020).
>
>
> [4] Rudi, Alessandro, Luigi Carratino, and Lorenzo Rosasco. "Falkon: An optimal large scale kernel method." Advances in Neural Information Processing Systems 30 (2017).
>
> [5] Charlier, Benjamin, et al. "Kernel operations on the GPU, with autodiff, without memory overflows." Journal of Machine Learning Research 22.74 (2021): 1--6.
>
> [6] Du, Simon S., et al. "Graph neural tangent kernel: Fusing graph neural networks with graph kernels." Advances in Neural Information Processing Systems 32 (2019).
>
> [7] Hron, Jiri, et al. "Infinite attention: NNGP and NTK for deep attention networks." International Conference on Machine Learning. PMLR, 2020.
>
> [8] Xiao, Lechao, Jeffrey Pennington, and Samuel Schoenholz. "Disentangling trainability and generalization in deep neural networks." International Conference on Machine Learning. PMLR, 2020.

---

### Review · Reviewer_UZ4u · 2025-04-14

**Summary Of Contributions:**

This paper introduces a novel representation called neural varifolds, which leverages deep neural networks and concepts from geometric measure theory to characterize the geometry of point clouds. The authors propose representing surfaces as distributions over positions and tangent spaces, combining spatial geometry with geometric consistency. Two algorithms for computing varifold norms using neural networks and their neural tangent kernel representations are presented. Extensive experimental results demonstrate the effectiveness of neural varifolds in shape matching and few-shot shape classification. It also has competitive performance in shape reconstruction tasks compared to state-of-the-art methods.

**Audience:**

Yes

**Claims And Evidence:**

Yes

**Requested Changes:**

**Questions**
- Have the authors explored methods to reduce the computational complexity of neural varifolds, such as kernel approximations?
- How robust is the proposed neural varifold method to noisy or incomplete normal information? Could the authors provide insights or experiments on the sensitivity of the proposed method?
- What motivated the choice of PointNet-based architectures, and would we have substantial performance gains from using more complex architectures (e.g., graph-based neural networks or transformers)?

**Strengths And Weaknesses:**

**Strengths**
- Integrating geometric measure theory with deep learning is novel and shows promise for robust geometric characterization.
- The authors provide clear mathematical foundations and practical algorithms, making the neural varifold concept well-grounded and applicable.
- The method is thoroughly validated across multiple tasks, demonstrating versatility and effectiveness, particularly in shape matching and few-shot classification scenarios.
- The thorough comparison of Chamfer distance, Earth Mover’s distance, etc., highlights the benefits and potential use-cases of neural varifolds clearly.

**Weaknesses**
- Kernel-based methods have quadratic complexity. The authors acknowledge this, but more practical methods or approximations to handle large-scale data effectively are not explored in depth.
- The neural varifold method relies heavily on normal vectors, which may limit its applicability in scenarios where accurate normal information is unavailable or hard to compute.
- The neural architecture used (PointNet-based) is relatively simple. The exploration of more advanced neural architectures (e.g., Graph CNNs or transformer-based models) is absent, potentially restricting performance improvements.

---

> ### Author Response · Authors · 2025-04-26
>
> **Weakness. Kernel-based methods have quadratic complexity. The authors acknowledge this, but more practical methods or approximations to handle large-scale data effectively are not explored in depth.**
>
> **Have the authors explored methods to reduce the computational complexity of neural varifolds, such as kernel approximations?**
>
> As we already mentioned in the main manuscript, the quadratic complexity is indeed a significant problem for kernel based approaches in the context of their applications to large datasets. Furthermore, in the context of neural tangent kernel theory, it is well known that even with exact kernel computation, infinite-width neural networks generally do not achieve classification accuracy on large datasets as high as their finite-width counterparts [1,2,3]. For example, in CIFAR-10 classification, the standard LeNet architecture significantly outperforms its NTK computation by a large margin and the computational costs for the standard neural network are significantly cheaper [1,2]. Thus, our study was mainly focused on the tasks where the kernel approach can compete with the the standard neural network approaches (e.g. shape similarity metrics, few-shot classification, and shape reconstruction).
>
> In the case of the few-shot classification (e.g. 1-shot, and 5-shot), kernel approximation might lose too much information, and thus we have not looked into the kernel approximation in details. In the context of the shape similarity measure, it would be very interesting to see how it accurately approximates the kernel pseudo-metric, but this requires more rigorous analysis. In the case of the shape reconstruction, Nystrom approximation could be used and makes the proposed algorithm more practical. Currently our method is based on neural tangent library, but it can be reimplemented to the FALKON [4] or KeOps [5], where more powerful GPU parallelization as well as kernel approximation methods are supported. As we are going to release our code upon the publication, we will support kernel based approximation as a follow-up.
>
> In the revision on page 10, we have added Section 5 for "limitation and conclusion.
>
>
> **Weakness. The neural varifold method relies heavily on normal vectors, which may limit its applicability in scenarios where accurate normal information is unavailable or hard to compute.**
>
> **How robust is the proposed neural varifold method to noisy or incomplete normal information? Could the authors provide insights or experiments on the sensitivity of the proposed method?**
>
> As the reviewer pointed out the surface normal information is often hard to compute or very noisy. For example, in the example of the shape reconstruction, we can compute the varifold kernel for the point clouds without any problem. But in the case of computing signed distances for each regular grid coordinates (e.g. 64 x 64 x 64), we need to assign arbitrary normal information since these are coordinates in the space by nature and they do not have surface normal information. As we wrote in Section Appendix A.2, we assigned the unit normal vector to the Z-axis to all these coordinates such that we can compute the varifold kernel representation between point clouds and regular grid coordinates, and thus compute the signed distance. As you can see in the result section, the reconstruction results are comparable to the popular shape reconstruction methods. As we pointed out in Section 4.3, this might be the main reason why the proposed method could not outperform the methods we compared in the manuscript in the case of shape reconstruction.

---

> ### Author Response · Authors · 2025-04-26
>
> **Weakness. The neural architecture used (PointNet-based) is relatively simple. The exploration of more advanced neural architectures (e.g., Graph CNNs or transformer-based models) is absent, potentially restricting performance improvements.**
>
> **What motivated the choice of PointNet-based architectures, and would we have substantial performance gains from using more complex architectures (e.g., graph-based neural networks or transformers)?**
>
> As the reviewer mentioned, the NTK form of more advanced architectures (e.g., graph neural network [6], and infinite-width attentions [7]) is available. However, unlike the PointNet architecture where its kernel computation is relatively simple, more advanced architectures require further computational optimization. Otherwise, their kernel form computation is in general very expensive. For example, graph neural network requires to compute the graph structure first and we need to store them, as well as propagates over the quadratic kernel representations. It requires additional complexity to the experiments. In particular, which graph neural network architecture we are going to choose, what the graph size is, and whether this is KNN graph or radius graph, etc.; there are many hyperparameters to control and optimize. In this study, we are proposing the neural varifold as a concept as well as a metric with relatively simpler hyper-parameterization; therefore, we choose relatively simple but efficient architecture (PointNet) to propose the concept and its potential applications. In the future study, we will explore the more advanced architectures with improved kernel computation as well as the pararellization with more kernel optimized software packages (e.g. KeOps [4], and FALKON [5]). Finally, we reassure that we will release the code and data used in the experiments for reproducibility.
>
> **references**
>
>
> [1] Arora, Sanjeev, et al. "On exact computation with an infinitely wide neural net." Advances in Neural Information Processing Systems 32 (2019).
>
> [2] Lee, Jaehoon, et al. "Finite versus infinite neural networks: an empirical study." Advances in Neural Information Processing Systems 33 (2020): 15156--15172.
>
> [3] Shankar, Vaishaal, et al. "Neural kernels without tangents." International Conference on Machine Learning. PMLR, 2020.
>
>
> [4] Rudi, Alessandro, Luigi Carratino, and Lorenzo Rosasco. "Falkon: An optimal large scale kernel method." Advances in Neural Information Processing Systems 30 (2017).
>
> [5] Charlier, Benjamin, et al. "Kernel operations on the GPU, with autodiff, without memory overflows." Journal of Machine Learning Research 22.74 (2021): 1--6.
>
> [6] Du, Simon S., et al. "Graph neural tangent kernel: Fusing graph neural networks with graph kernels." Advances in Neural Information Processing Systems 32 (2019).
>
> [7] Hron, Jiri, et al. "Infinite attention: NNGP and NTK for deep attention networks." International Conference on Machine Learning. PMLR, 2020.

---

### Author Response · Authors · 2025-04-09

Thank you to all reviewers for their careful reading and feedback. We are currently working on revising the manuscript and addressing the weaknesses and requested changes highlighted by the reviewers. We look forward to the discussion with the reviewers.

---

### Comment · Editors_In_Chief · 2025-07-04

By request of the authors, on July 1, 2025, the Editors-in-Chief replaced the paper's PDF after it was published. Changes are largely grammatical changes, typo fixes, and the removal of a definition that was not needed.

---

### Decision · Action_Editor_JsUj · 2025-05-20

**Recommendation:** Accept with minor revision

**Comment:**

The reviewers all acknowledged the contributions and value of this work. They nonetheless expressed come concerns, particularly regarding the computational complexity of the method and the use of relatively simple architectures. The authors addressed these points in their responses and commented on them in the paper. Ultimately, all reviewers recommended acceptance in their final recommendation.

Although the AE believes that most of the important points raised in the reviews were incorporated in the revised version of the manuscript, it seems that it was not the case for the question of Reviewer qt59 on sensitivity to hyper-parameters. As the discussion of hyper-parameter values is important for reproducibility, the AE requests the authors to discuss this in the final version of the paper.

**Audience:**

The reviewers all agree that there is a TMLR audience for this work.

**Claims And Evidence:**

The reviewers all acknowledge that the claims are supported by sufficient evidence.